# DustNet (v1): Skillful neural network predictions of dust aerosols over the Saharan Desert

Trish E. Nowak[1,3], Andy T. Augousti[2], Benno I. Simmons[3,*], and Stefan Siegert[1,*]

[1]Mathematics and Statistics, University of Exeter, Exeter, EX4 QH, UK
[2]Department of Mechanical Engineering, Kingston University, London, SW15 3DW, UK
[3]Centre for Ecology and Conservation, University of Exeter, Penryn, TR10 9FE, UK
[*]These authors should be considered joint last author.

**Correspondence:** Trish E. Nowak (pn284@exeter.ac.uk) and Stefan Siegert (s.siegert@exeter.ac.uk)

**Abstract.** Suspended in the atmosphere are millions of tonnes of mineral dust that interact with weather and climate. Accurate representation of mineral dust in weather models is vital, yet it remains challenging. Large-scale weather models use supercomputers and take hours to complete forecasts. Such computational burdens allow them to include only monthly climatological means of mineral dust as input states, inhibiting their forecasting accuracy. Here, we introduce DustNet, a simple, accurate, and fast forecasting model for 24-hour (1-step) ahead predictions of aerosol optical depth (AOD). DustNet is a custom-built 2-D Convolutional Neural Network (CNN) equipped with transposed convolution layers. The model is trained on selected ERA5 meteorology and past MODIS-AOD observational data as inputs. Our design of DustNet ensures that the model trains in less than 8 minutes and creates predictions in 2.1 seconds on a desktop computer, without the need to utilise any Graphics Processing Units (GPUs). Predictions created by DustNet outperform the state-of-the-art physics-based model at coarse $1°$ x $1°$ resolution at $95\%$ of grid locations when compared to ground truth satellite data. The test results show that the daily mean AOD over the entire Saharan Desert area highly correlates with MODIS observational data, with Pearson's $r^2 = 0.91$. Our results demonstrate DustNet's potential for fast and accurate AOD forecasting, which can easily be utilised by researchers without access to supercomputers or GPUs.

## 1 Introduction

The Earth's atmosphere is loaded with approximately 26 million tonnes of mineral dust - an atmospheric aerosol that represents the vast majority of mass burden in the atmosphere (Gliß et al., 2021; Kok et al., 2023). Each year, major sources emit nearly 5,000 million tonnes of dust globally (Kok et al., 2021b) and, although the majority of this material sinks at source, a substantial portion is transported over vast distances (Van Der Does et al., 2018). Once in the atmosphere, mineral dust interacts with the Earth systems and impacts weather, climate, human health and infrastructure, from fisheries to aviation (Shao et al., 2011;

Knippertz and Stuut, 2014; Highwood and Ryder, 2014; Nenes et al., 2014; Miller et al., 2014; Jickells et al., 2014; Morman and Plumlee, 2014; Kok et al., 2023).

Despite its importance, representing atmospheric dust aerosols in weather and climate models is challenging (Parajuli et al., 2022; Kok et al., 2023). For example, physics-based Numerical Weather Prediction (NWP) and climate models struggle to fully represent the dust cycle with adequate emission, transport and generation (Evan et al., 2014; Kok et al., 2021a; Gliß et al., 2021; Zhao et al., 2022). Instead, the Integrated Forecasting System (IFS) of the European Center for Medium-Range Weather Forecasting (ECMWF) creates predictions that use aerosol optical depth (AOD) based on monthly-mean climatological fields only (Bozzo et al., 2017). A limitation in computational resources is highlighted as one of the reasons for the lack of a dedicated aerosol scheme, since such a development would significantly increase the computational burden of the system (Mulcahy et al., 2014). The monthly mean AOD, developed by the Copernicus Atmosphere Monitoring Service (CAMS), provides a reasonable trade-off in global weather forecasting. However, a more accurate representation of the AOD would have significant benefits, such as large improvements in the representation of the summer monsoon circulation or precipitation patterns in the Sahel region (Bozzo et al., 2020; Balkanski et al., 2021).

Recent developments in the field of AI present a significant opportunity to overcome the computational burden of a dedicated physics-based aerosol scheme. Models such as GraphCast, Pangu-Weather, and FourCastNet can now skillfully predict the main ERA5 variables and in many cases outperform the state-of-the-art NWP models (Lam et al., 2023; Bi et al., 2023; Pathak et al., 2022). To date, attempts to forecast atmospheric aerosols with neural network architectures have shown varying levels of success. "Satisfying" results were reported (Kang et al., 2019; Daoud et al., 2021) when applying a long-short-term memory (LSTM) architecture to local AOD forecasts. The application of a U-NET architecture revealed a skillful detection of classified 'dust events' at 67 % precision rate (Sarafian et al., 2023). A lack of comparisons to the current physics-based forecasts, or inclusion of standardised skill metrics, makes direct comparison between AOD forecasting models nearly impossible.

Here, we present a unique application of 2D convolutional neural networks (CNN) to forecast atmospheric aerosol levels. We use our model (hereafter 'DustNet') to produce 24-hour (1-step) spatial forecasts of AOD over North Africa. Computationally cheap DustNet runs on a modestly configured laptop, rather than a high-power computer (HPC) - a fraction of the computational power required by traditional NWP models. The model trains in less than 8 minutes and predicts in 2.1 seconds. We compare the predictions of DustNet, and the corresponding daily CAMS forecasts, against the satellite-derived data using standard evaluation metrics, such as the root mean squared error (RMSE) and an accuracy correlation coefficient, to facilitate easy comparison with future AI models. The advantage of a smaller processing power requirement and rapid speed of prediction, combined with the accuracy of the forecast, makes our model a valuable complement to traditional AOD forecasting systems.

## 2 Methods

### 2.1 Study area

To effectively forecast dust aerosols, our study area encompasses the global principal dust generation source - the Sahara Desert, which is responsible for over 55% of the 1,536 million tonnes of total global dust emitted annually (Ginoux et al.,

2012). The region covers an area from 20°W - 31°E and 0° - 31°N (51 × 31 grid cells), with a longitudinal centre around the Bodélé Depression (16.5°N, 16.5°E). Located in northern Chad, this single location generates an estimated 6–18% of global dust emissions, which total to approximately $182(\pm65)$ million tonnes per year, the region is of major importance in models that seek to capture dust generation (Todd et al., 2007). To capture the seasonal south-westward dust transport across the Sahara and towards the Atlantic Ocean, our region includes additional grid cells to the South and West of the Bodélé Depression.

This choice allowed us to gain a sufficient amount of training data, with $51 \times 31$ grid cells providing 1,581 pixels for each training day, thereby ensuring robust model performance. By selecting this region, we were able to strike a balance between training efficiency, training speed, and prediction accuracy, making it possible to achieve effective dust aerosol forecasting. Furthermore, this approach enabled us to train the model on a traditional desktop computer without relying on cloud resources for data storage, making our approach more accessible and cost-effective. Additionally, the study region effectively captures

dust aerosol generation and transport on selected features, which is essential for accurate forecasting. Finally, by minimizing the area to the Saharan Desert and consequently reducing the amount of chosen training features, we were able to avoid adding different ocean and terrain processes, leading to reduced model complexity without compromising performance.

## 2.2   Datasets

### 2.2.1   AOD data

We retrieved the AOD data from the Moderate Resolution Imaging Spectroradiometer (MODIS) instrument located on board both Aqua and Terra spacecraft. With daily temporal resolution over a period of 20 years starting from 1st January 2003 to 31st December 2022, the AOD data yields $2 \times 7305$ files. We used the quality-controlled Level-3 data for AOD at 550nm. Choosing the combined mean of Dark Target and Deep Blue algorithms provided a full coverage above bright and dark surfaces at a horizontal resolution of $1° \times 1°$ (Hubanks et al., 2015). This choice provided a good spatiotemporal coverage of AOD data

above both land and ocean surfaces.

### 2.2.2   ERA5 data

Meteorological data comes from the fifth generation of European Centre for Medium-Range Weather Forecast (ECMWF) Atmospheric Reanalysis Project (ERA5) and consists of 5 parameters: wind u component, wind v component, vertical velocity, temperature and relative humidity. Each parameter was retrieved at 5 pressure levels 550hPa, 750hPa, 850hPa, 950hPa and

1000hPa. This choice provided us with 35 distinctive features representing atmospheric conditions from ground level to $\approx 5$km in vertical height. The ERA5 data is available on an hourly basis, but here we only chose the data representing conditions for midday (12:00 UTC). This allows us to represent the mid-point in atmospheric conditions between the Terra and Aqua satellite overpasses above the equator (10:30am and 1:30pm respectively). To further match the meteorological data with AOD, we chose a daily temporal resolution between 2003-2022. The horizontal resolution of ERA5 data is $0.25° \times 0.25°$. To match this

with the AOD resolution of $1° \times 1°$, the data was regridded (see section Data pre-processing for details).

### 2.2.3 Timestamps

We created timestamps using the NumPy package (version 1.23.0) in Python with a daily temporal resolution over 20 years from 2003 to 2022 (7,305 days). We then expanded the array dimensions through replication to match the exact spatial resolution of atmospheric variables, resulting in a coverage of $31 \times 51$ grid cells for each day.

### 2.2.4 Elevation

We obtained global elevation data at a resolution of $1° \times 1°$ from the Joint Institute for the Study of the Atmosphere and Ocean at the University of Washington (Mitchell, T., 2014) and extracted grid locations for our study area. Similar to the timestamp data (see 2.2.3), we expanded the terrain array's dimensions to match the temporal resolution of the atmospheric variables. This was achieved through replication, resulting in an array shape of $7,305 \times 31 \times 51 \times 1$.

### 2.2.5 CAMS forecast

We obtained daily 'Total aerosol optical depth at 550nm' forecast data from 'CAMS global atmospheric composition forecasts'. CAMS forms a part of the ECMWF Integrated Forecasting System (IFS), and is a sophisticated numerical weather forecasting model (NWP) (Bozzo et al., 2017). During the AOD data assimilation process, CAMS utilises data from MODIS, among other satellites, together with data from ground-based observation stations. The model then uses physics and chemistry principles to forecast hourly AOD values on a single level for up to 5 days (120 h) ahead (Morcrette et al., 2009; Benedetti et al., 2009). For consistency, we only chose forecasts representing 12:00 UTC to capture the midpoint conditions between Aqua and Terra overpasses above the equator. The temporal extent choice was also matched to our predictions. Therefore, we initiated forecasts on midday 1st January 2020 until 30th December 2022 for 1095 days forecast between 2nd January 2020 and 31st December 2022. CAMS data is provided at a $0.4° \times 0.4°$ spatial resolution. To match with our data, we therefore used an identical approach as for the ERA5 datasets to regrid to a $1° \times 1°$ resolution (details in Data pre-processing section).

## 2.3 Data pre-processing

### 2.3.1 Data imputation

We combined data from the MODIS Aqua and Terra data sources at each individual location and time by labelling AOD data as missing whenever both sources were missing, using available data from one source if the other is missing, and averaging both sources whenever both are available. This data combination step reduces the total fraction of missing AOD values from $32.81\%$ in Aqua and $30.89\%$ in Terra to $19.89\%$ in the combined data set. The remaining missing AOD values are imputed by spatial interpolation (individually for each time step) using Lattice Kriging (Hartman and Hössjer, 2008; Rue and Held, 2005) on four nearest neighbours with uniform weights. To validate the imputation method, we randomly held out $10\%$ of the AOD data and compared them to their imputed values. The mean squared error of the imputed values is 0.005 which is less than $5.30\%$ of the total variance of the AOD data. The MSE was found to be insensitive to the choice of the Kriging hyperparameter,

with relative differences of less than $0.0003\%$ over a wide range of values (see Supplementary Fig. S1). See Section 'Code and data availability' for links containing the pre-processed data and full Python code for imputation.

### 2.3.2 ERA5 regridding

The ERA5 data (Hersbach et al., 2018) is supplied with a horizontal resolution of $0.25° \times 0.25°$ and thus needed regridding to match the AOD resolution. We processed all meteorological data using Python version 3.8.13 and the Iris v 3.2.1 package. We used nearest-neighbour interpolation from the Iris package to convert each feature to a common $1° \times 1°$ resolution.

### 2.3.3 Feature engineering

To enhance the model's predictive skill we incorporated two aspects of feature engineering: AOD lag and seasonal features. To account for temporal dependences, we use 5 preceding days of AOD data as features to predict AOD on a given day. Hence, we had to remove the first 5 timestamps form the database as these did not have complete features available, consequently reducing the total number of timestamps to 7,300. Additionally, we included trigonometric transformations of timestamps as seasonal features using sine:

$$x_{ijt}^{(42)} = \sin\left(2\pi \frac{t}{365.2425}\right), \tag{1}$$

and similarly using the cosine:

$$x_{ijt}^{(43)} = \cos\left(2\pi \frac{t}{365.2425}\right), \tag{2}$$

where $t$ represents the day of the year. Timestamps are constant across space and allow the model to represent periodic variations on seasonal timescales. Thus, together with timestamps, our final total input consisted of 43 features.

### 2.3.4 Combining and normalising

We combined the meteorological data with AOD data into a single 4D NumPy array of shape 7300, 51, 31, 43, where the first dimension represents time, the second and third are longitude and latitude respectively, and features are stored along the last dimension. Let the $x_{ijt}$ be the value of feature $x$ at grid point $i, j$ and time $t$. We normalised all features using min-max normalisation:

$$x_{ijt,norm} = \frac{(x_{ijt} - x_{min})}{(x_{max} - x_{min})} \tag{3}$$

where $x_{min}$ and $x_{max}$ are the overall minimum and maximum of a feature $x$ over all grid points and timestamps in the training data.

### 2.3.5 Training, validation, test split of data

We split the data along the time dimension into $70\%$, $15\%$ and $15\%$ for training, validation and test sets respectively. Splitting data with consecutive time steps yielded better results than a random split. Therefore, the training set covered 5,110 consecutive

days from 6th January 2003 until 1st January 2017 (inclusive of both days). The use of consecutive time steps ensures that each subset is composed of data points that are temporally distinct. This method reduces the risk of autocorrelation and improves the model's ability to generalize to new, unseen data (Rasp et al., 2020). The validation set took 1,095 consecutive days from 2nd January 2017 to 1st January 2020. Finally, we set aside a test set, with 1,095 days of data from 2nd January 2020 to 31st December 2022. We made sure that the model never had access to the test set during the training and validation processes and only after these were complete did we introduce the test data and run our model to obtain predictions. All pre-processed data and code are available for download from a public repository (see Section 'Code and data availability' for links to both data and code).

## 2.4 Designing CNN models

To find the best forecast of the daily AOD, we designed three CNN models based on Hinton et al. (1995); LeCun et al. (2015) and Goroshin et al. (2015). We used the end-to-end open source machine learning platform TensorFlow 2, together with the Keras high-level API (Abadi et al., 2016; Chollet et al., 2015). Each model uses a different architecture based on two-dimensional (2D) convolutions (hereafter Conv2D). In general, the Conv2D neural network architecture enables regression problems in image analysis to be addressed and is particularly effective at capturing spatial patterns in two-dimensional images. The efficiency of Tensorflow allows for training and inference to be run on traditional desktops or laptops rather than requiring HPC's. All models described hereafter were run using Python version 3.10.10 on a MacBook Pro with an Apple M1 Pro and 32GB RAM. Since the models did not use any GPUs, they can be easily replicated by users without an access to a supercomputer.

We have chosen 'Adam' optimizer and the mean squared error (MSE) as a loss function. These options offered optimal results in terms of training times and were used for further analysis. For the Adam optimizer we used a learning rate of 0.001 and an exponential decay rate of 0.9, which are default settings following Kingma and Ba (2014).

We determined the optimal size of the convolving window (kernel size) and the number of strides with a series of diagnostic tests. The results of these tests are presented in Table 1 with the optimal choice in bold based on minimising the mean squared error and the speed of the training time. The final design included a kernel size of (2,2) with a stride equal to 2, which produced the optimal MSE to training time ratio. We recognise that we have not tested every possible combination, thus it may be possible to achieve a better performing design. Python codes for all three models with accompanying training data are available for download from a public repository (see Data and Code availability section for links).

We initially assigned 50 epochs to each training regime and monitored the performance using the mean squared error of training to validation loss. We also configured each model with Early Stopping and a patience of 4 epochs. This set up halts the training time when there is no improvement in validation loss after 4 consecutive iterations and prevents the model from over-fitting to training data (see Supplementary Fig. S2). Our setup saved the optimal ratio of training time versus validation loss and used the best performance to run predictions. Below, each model's architecture is described in detail.

**Table 1.** Effects of choosing different kernel sizes on training time and MSE for 2 models: Conv2D and U-NET. For simplicity, this test was run on a subset of data. The optimal choice is presented in bold font. Note that a small improvement in the MSE for a kernel size (3,3) was disregarded in favour of a much faster training time and time per step for kernel size (2,2).

| Kernel size | Conv2D | | | U-NET | | |
|---|---|---|---|---|---|---|
| | (5,5) | (3,3) | (2,2) | (5,5) | (3,3) | (2,2) |
| Training time | 42min | 23min | **3min** | 1h41min | 1h7min | **23min** |
| Time per step | 28s | 28s | **12s** | 144s | 140s | **88s** |
| MSE | 0.00174 | 0.00133 | **0.00134** | 0.00175 | 0.00148 | **0.00151** |

### 2.4.1 Conv2D model

For the first AOD prediction model we adapted a classical design of CNN. The Conv2D architecture, inspired by the visual system, applies filters (or convolutions) to capture spatial patterns in two-dimensional images (LeCun et al., 2015). The network performs feature extraction and learns representations at different scales. Such representations allow the network to identify relevant information and thus make predictions. Learning of the complex representation is made possible by the non-linearity provided to the model by a correctly chosen activation function. Ramachandran et al. (2017) suggested an improvement to the popular Rectified Linear Unit or 'ReLU' activation function (Agarap, 2018; Nair and Hinton, 2010) by proposing the Swish activation function. This method gained in popularity as it is capable of smoother output representation as well as more consistent performance (Rasamoelina et al., 2020). Since the Swish activation function proved to yield the best performance, we used it with all 5 hidden layers. Each hidden layer in our Conv2D model was designed with a maximum of 264 and a minimum of 16 filters, as well as a $2 \times 2$ kernel size, which specifies the height and width of the 2D convolution window (see Fig. A1 for model sketch). The final output convolution used a single hidden layer with the 'ReLU' activation function. An architecture constructed in this way provided 218,673 trainable parameters.

### 2.4.2 U-NET model

The architecture of our second model employed a U-NET like design, first proposed by Ronneberger et al. (2015) for the purpose of biomedical image classification. The model is characterised by its "U" shape design which employs both contracting and expanding pathways to identify specific features within images. Here, we follow the approach of Ayzel et al. (2020) who, inspired by U-NET, designed their RainNet model for precipitation nowcasting. Thus, we also divided our model into two parts, encoder and decoder, and utilised skip connections between both paths via concatenation layers - unique features of the U-NET model. The U-NET model design sketch can be found in Fig. A2. The encoder (or contracting) pathway of the model included six Conv2D layers with Swish activation and a $2 \times 2$ kernel size, as well as two MaxPooling2D layers with pool size $2 \times 2$. The decoder (or expanding) pathway had five Conv2D layers with two UpSampling2D and two Concatenate layers. The input layers were bordered with a ZeroPadding2D layer which was cropped to the original size of $31 \times 51$ with Cropping2D in

the output layer. Unlike the original U-NET network, our design received 4-dimensional arrays of shape $7,300 \times 31 \times 51 \times 43$ and generated an output image of a shape of $31 \times 51 \times 1$ for each prediction time step. The final U-NET model architecture provided 847,937 trainable parameters.

### 2.4.3 DustNet model

The last model design built upon the architecture of Conv2D and U-NET. This unique design replaces the Concatenation layers with Transpose convolution layers, also known as Deconvolutional Networks (Zeiler et al., 2010). Schematically represented in Fig. 1 the input layer was first padded with a border of zeros (ZeroPadding2D), which increased the input shape from $31 \times 51 \times 43$ to $40 \times 64 \times 43$. Zero padding enabled the convolution to produce the same output size for multiple input sizes (Dumoulin and Visin, 2016). We then applied the 2D convolving windows (Fig. 1 - pink arrows) which moved over each padded input with a $2 \times 2$ kernel size and $2 \times 2$ strides that allow upsampling. The first six layers of convolving, or contracting pathway, consisted of double 64, 128 and 256 filters, where every second layer included strides. This allowed the model to decrease the input size while increasing the number of channels ($5 \times 8 \times 256$). The 'deconvolution', or expanding pathways, were then applied by adding six Conv2D Transpose layers, with reversed order of filters to the contracting pathway. An advantage of transposed convolution is its ability to efficiently upscale input data by applying inverse convolutions. This enables the network to increase the size compared to the input and thus generates high-resolution images at finer spatial scales (Zeiler et al., 2010). A 2D cropping layer was then added to bring the width and height back to its initial input size of $31 \times 51$, while the final convolution with a single filter matched the output with the desired target size of $31 \times 51 \times 1$. This architectural design allowed the model to create a total of 1,291,009 trainable parameters.

### 2.4.4 Baseline models

We set the baselines as AOD climatological mean and persistence. The climatological means were calculated separately at each spatial location as the mean AOD over the training period. The climatological benchmark is constant in time. A time-varying baseline model is the persistence forecast, which uses the most recent observation of AOD as the 24-hour ahead prediction. Here, we used the values from the 1st day of calculated AOD lag from the reserved test set (values unseen by the model) to represent persistence. Both climatology and persistence act as null models, and a more sophisticated forecasting scheme should be able to outperform both in order to be considered useful.

### 2.5 Training CNN models

To train the models we used 17 years of daily data (2003-2019). We initiated the training on the first 15 years (70%) of data after which the models went into a self validation mode, for which we used the consecutive 2 years (15%) of data (see Section 2.3.5 for full details on data split regime). The inputs included the value of the AOD over the previous 5 days and previous 1 day for each of 35 meteorological features (7 atmospheric variables at 5 pressure levels, see ERA5 data section in Methods). Regridded to a $1°\text{x}1°$ resolution over $31°$ of latitude by $51°$ of longitude, together with orography and the sine and cosine

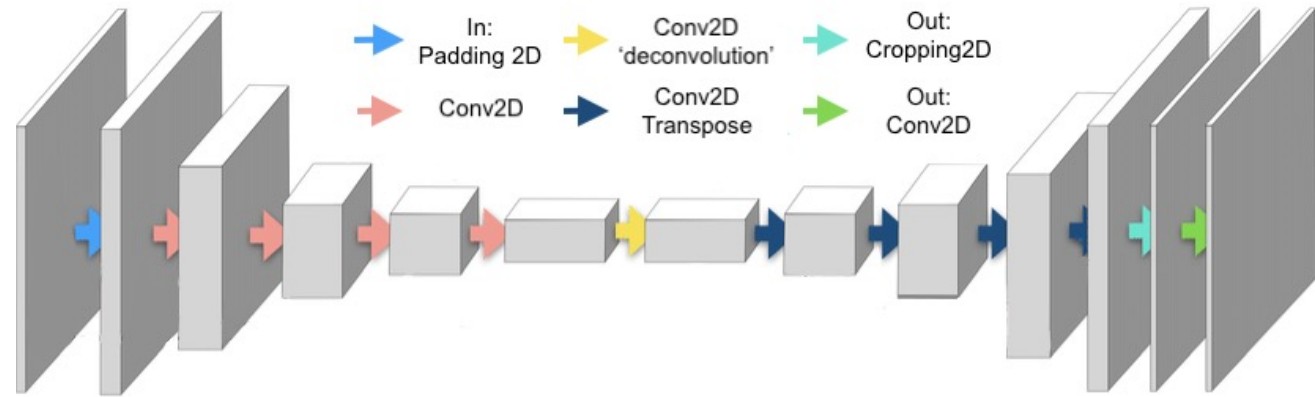

**Figure 1.** Schematic representation of the DustNet model. Each of 6,205 inputs is first padded with a border of zeros using ZeroPadding2D (light blue arrow) to increase input shape and allow the convolution windows to detect the borders. The features are then extracted by 2D convolution window (pink arrows) which decreases input shape while increasing the number of trainable parameters. Then deconvolution is applied (yellow arrow) by including a 2D transpose network, which increases the size of the input (dark blue arrows) while maintaining connectivity between the layers. The output is then cropped back to match the initial input size (cyan arrow) and sent through a final 2D convolution (green arrow) to produce a 24-hours (1-step) ahead prediction.

values of timestamps, the data produced a representative state consisting of 43 input features. Hence, for each of the 6,205 training and validation days the models had access to 67,983 values.

### 2.6 Statistical analysis

#### 2.6.1 Evaluation of CNN models

To evaluate the 24-hour ahead (1-step) predictions, we used 3 years of daily data (2020-2022), which were unseen by the
235 models. Our initial baseline model included the climatological mean, which is often used in meteorological forecasts as a sensible default (Bozzo et al., 2020). We evaluated each CNN model's performance by assessing the training time, inference time taken per 'time-step', the MSE of predicted values in the test set, and the percentage improvement in the MSE above the climatology and persistence baseline models. We then used the best performing model to visually evaluate its output against (unimputed) MODIS values. We initially inspected model's daily predictions for its ability to represent AOD spatially by
240 mapping 28 consecutive days of predictions next to the corresponding data from MODIS (see Supplementary Fig. S3). We looked for the model's ability to capture the main dust generation sources, consistent AOD transport with prevailing winds, and correct distinctions of AOD accumulation between the ocean and land border.

To analyse the errors of the best performing model, we rearranged Equation 3 reverses normalisation of AOD predictions from each model:

$$y_{ijt,denorm} = y_{ijt,pred}\left(y_{max} - y_{min}\right) + y_{min},$$ (4)

where $y_{pred}$ are the values predicted by the model, $y_{max}$ is the maximum and $y_{min}$ is the minimum AOD value from the training set. In this same manner, we used Equation 4 to reverse normalisation of the climatology and persistence predictions. We then assessed each CNN model by calculating the MSE between values predicted by the model using the de-normalised AOD denoted as $\hat{A}$, and the corresponding values from the test set ("true") AOD value devoted as $A$. Here, we calculated a mean value along an axis of latitude $N_{lat}$ and longitude $N_{lon}$, of our spatial coordinates at each prediction time step $t$, where $N_{lat}$=31, $N_{lon}$=51 and $N_t$=1095, using Equation 5:

$$MSE = \frac{1}{N_{lat}N_{lon}N_t}\sum_{i=1}^{N_{lat}}\sum_{j=1}^{N_{lon}}\sum_{t=1}^{N_t}(\hat{A}_{ijt} - A_{ijt})^2,$$ (5)

We used this same process as described above to obtain the MSE for the climatology and persistence models. To ensure that model evaluation is only based on actually observed AOD values, all imputed AOD values were excluded from calculation of the MSE.

### 2.6.2 Validation of results

To validate our results, we compared our predictions with the ground-truth (not imputed) data from MODIS and the physics-based model (CAMS) fairly, we calculated the following metrics: mean bias error (MBE), RMSE, difference between RMSE's ($\Delta$RMSE) and ACC. The metrics, defined below, follow a combination of notations from Bi et al. (2023) and Lam et al. (2023) adapted to spatial representation of temporally averaged values for each prediction day $t$ ($N_t$=1095). All prediction values were first de-normalised using Equation 4. Subsequently, we compared the model predictions ($\hat{A}$) with raw (unimputed) MODIS data (mean of Aqua and Terra) denoted as $A$. The climatological mean, denoted as $A'$, corresponds to the long-term average of AOD values from MODIS (2003-2022). To allow for comparison with the physics-based forecast, we tested the 24-hour (1-step) ahead predictions from CAMS using these same skill metrics, and compared them with the daily and seasonal results produced by the best performing model.

### 2.6.3 Spatial analysis

To analyse spatial characteristics of model performance, we calculated the temporal mean of model predictions ($N_t$ = 1095) at each location (lat, lon). This allowed us to calculate mean bias error (MBE) between the predicted AOD ($\hat{A}$) and MODIS ground truth ($A$) for both, the best performing model and CAMS, using Equation 6.

$$MBE_{spatial,ij} = \frac{1}{N_t}\sum_{t=1}^{N_t}(\hat{A}_{ijt} - A_{ijt})$$ (6)

We also calculated the spatial root mean square error ($RMSE_{spatial}$) for each model using Equation 7.

$$RMSE_{spatial,ij} = \sqrt{\frac{1}{N_t}\sum_{t=1}^{N_t}(\hat{A}_{ijt} - A_{ijt})^2} \tag{7}$$

Calculating differences between RMSEs (ΔRMSE) using Equation 8 allowed us to reveal specific locations at which predictions from one model outperformed the other.

$$\Delta RMSE_{spatial,ij} = RMSE_{spatial,ij}^{(CAMS)} - RMSE_{spatial,ij}^{(CNN)} \tag{8}$$

Additionally, we calculated the spatial distribution of Anomaly Correlation Coefficient (ACC, Equation 9). Let $\hat{A}'$ be the anomaly of predicted AOD values ($\hat{A}$), and $A'$ the anomaly of observed (ground truth $A$) AOD values, where the anomalies are the differences from MODIS climatology values, then:

$$ACC_{spatial,ij} = \frac{\sum_{t=1}^{N_t}\left[(\hat{A}'_{ijt} - \bar{A}'_{ijt}) \times (A'_{ijt} - \bar{A}'_{ijt})\right]}{\sqrt{\left[\sum_{t=1}^{N_t}(\hat{A}'_{ijt} - \bar{A}'_{ijt})^2\right] \times \left[\sum_{t=1}^{N_t}(A'_{ijt} - \bar{A}'_{ijt})^2\right]}} \tag{9}$$

The ACC is a common measure of skill which assesses the quality of prediction, and highlights anomalies between forecast and observed values. By subtracting the climatological mean from both, prediction and verification, the ACC measures the quality of prediction without giving misleadingly high results caused by seasonal variations.

### 2.6.4 Temporal analysis

To analyse the model's predictions across different times, we calculated mean spatial AOD values for each prediction day. We also computed Pearson's correlation coefficients (r), associated p-values, and coefficient of determination ($r^2$) using the SciPy statistical package v.1.12 for each prediction day (N = 1095) of spatially averaged data ($N_{lat}$, $N_{lon}$ = 31, 51). Corresponding calculations were performed for both, the best performing model and CAMS forecasts, with the MODIS ground truth data. We have also adapted Equations 6 and 7 to temporal representation by using Equation 10 and 11.

$$MBE_{temporal,t} = \frac{1}{N_{lat}N_{lon}}\sum_{i=1}^{N_{lat}}\sum_{j=1}^{N_{lon}}(\hat{A}_{ijt} - A_{ijt}) \tag{10}$$

$$RMSE_{temporal,t} = \sqrt{\frac{1}{N_{lat}N_{lon}}\sum_{i=1}^{N_{lat}}\sum_{j=1}^{N_{lon}}(\hat{A}_{ijt} - A_{ijt})^2} \tag{11}$$

### 2.6.5 Justification of the selected points

In addition to spatial and temporal analyses, we focussed on four point locations to assess the model's performance at the local scale. The locations, shown in Fig. 2, were selected on the basis of a different aerosol type contributing to the total AOD, as well as prevailing meteorological conditions. We chose the region around the Bodélé Depression in Chad (16.5°N, 16.5°E) for its

dust generation capability and consistency of high mineral dust loading (Washington et al., 2003). Nouadhibou in Mauritania (20.5°N, 17°W) is located at the edge of western Africa, where hot and dry Saharan air meets cool and moist Atlantic air (Carlson and Prospero, 1972). The temperature inversion creates a barrier for low horizontal flow of atmospheric dust, and instead forces an uplift of over 1.5km (Prospero and Carlson, 1972). From this point atmospheric dust moves westward towards Central and South America at higher altitudes between 1.5km - 5km (Kaufman et al., 2005). To capture the transport of dust and fire smoke with southwestward winds towards South America (Kaufman et al., 2005) we chose a location over the Atlantic Ocean in the Gulf of Guinea (4°N, 4°W). For the fourth location, we chose the second largest city in Nigeria and the capital of Kano State (11.5°N, 8.5°E). Kano City is on a direct pathway of seasonal dust plumes known locally as the Harmattan season. During boreal winter the wind direction shifts to southwestward direction and transports the sand storms generated from the Bodélé Depression towards Kano, where they are associated with a large increase in air pollution (Anuforom, 2007; Schwanghart and Schütt, 2008; Sunnu et al., 2008).

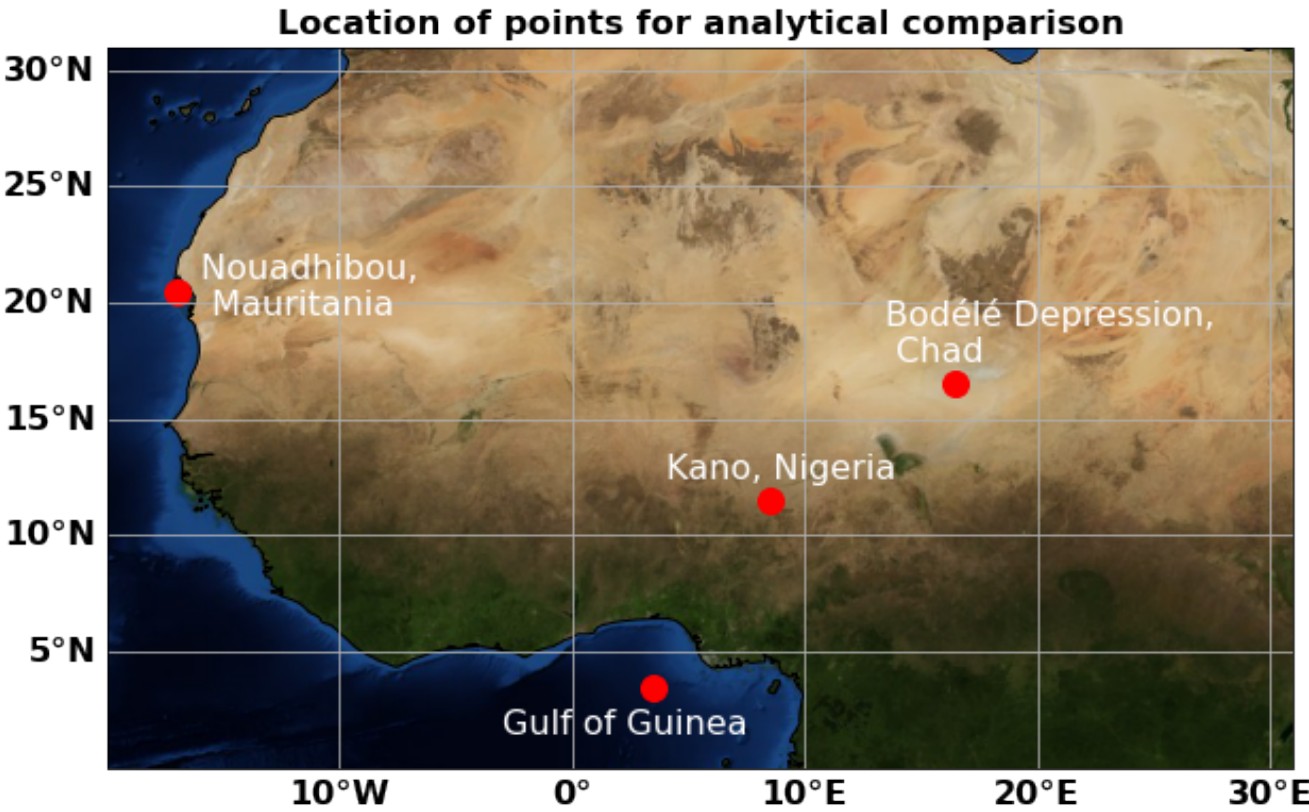

**Figure 2.** Study area and the locations of selected grid points used to asses the model's predictive accuracy on a local scale ($1° \times 1°$ resolution). The background image for the December view of Blue Marble is available from NASA at https://visibleearth.nasa.gov/collection/1484/blue-marble?page=4.

### 2.6.6 Feature importance

We assessed feature importance using a perturbation-based method, where individual input channels were systematically altered to evaluate their contribution to model predictions. Specifically, each feature was zeroed out in turn, and the mean squared error (MSE) between the full prediction and the prediction with the altered input was calculated. This approach quantifies the sensitivity of the model's output to the absence of each feature, with higher MSE indicating greater importance. Perturbation-based methods, such as this one, are widely used for assessing feature relevance in machine learning models due to their simplicity and interpretability (Covert et al., 2021; Molnar, 2022).

## 3 Results

### 3.1 Performance verification

The comparative results of the three CNN models, shown in Table 2, demonstrate a clear advantage of the DustNet architecture in both computational efficiency and predictive accuracy. Developed in this study, DustNet achieves the shortest training time at 7 minutes and 41 seconds, over a third that of the U-NET model. It also outperforms both U-NET and Conv2D in terms of mean squared error (MSE), achieving a value of 0.00153, which corresponds to a 53.68% improvement over the climatology baseline. Furthermore, DustNet generates forecasts in just 2.1 seconds, making it the fastest among the tested models. In contrast, Conv2D and U-NET require over 13 and 25 minutes, respectively, for training, while their resulting predictions show lower improvement over the climatological baseline. These findings highlight that DustNet is both more efficient and more accurate than the Conv2D and U-NET models, thereby demonstrating its skill in deterministic AOD forecasting.

**Table 2.** Normalised test results for three unique model architectures. Climatology and persistence baseline MSE's of prediction to test data are presented below the table. The rows display results for total training time, time per iteration step and MSE for each kernel size of each model. The last column shows the percentage difference when compared to the climatological baseline.

| CNN model | Training time | Time per step | MSE | Prediction time | Baseline[1] improvement (%) |
|---|---|---|---|---|---|
| Conv2D | 13min40s | 34s | 0.001895 | 4.1s | 42.63% |
| U-NET | 25min20s | 53s | 0.001691 | 4.9s | 48.80% |
| DustNet | **7min41s** | **17s** | **0.00153** | **2.1s** | **53.68%** |

Baseline MSE:
[1] Climatology: 0.003303
[2] Persistence: 0.002992

### 3.2 Performance of spatial forecast

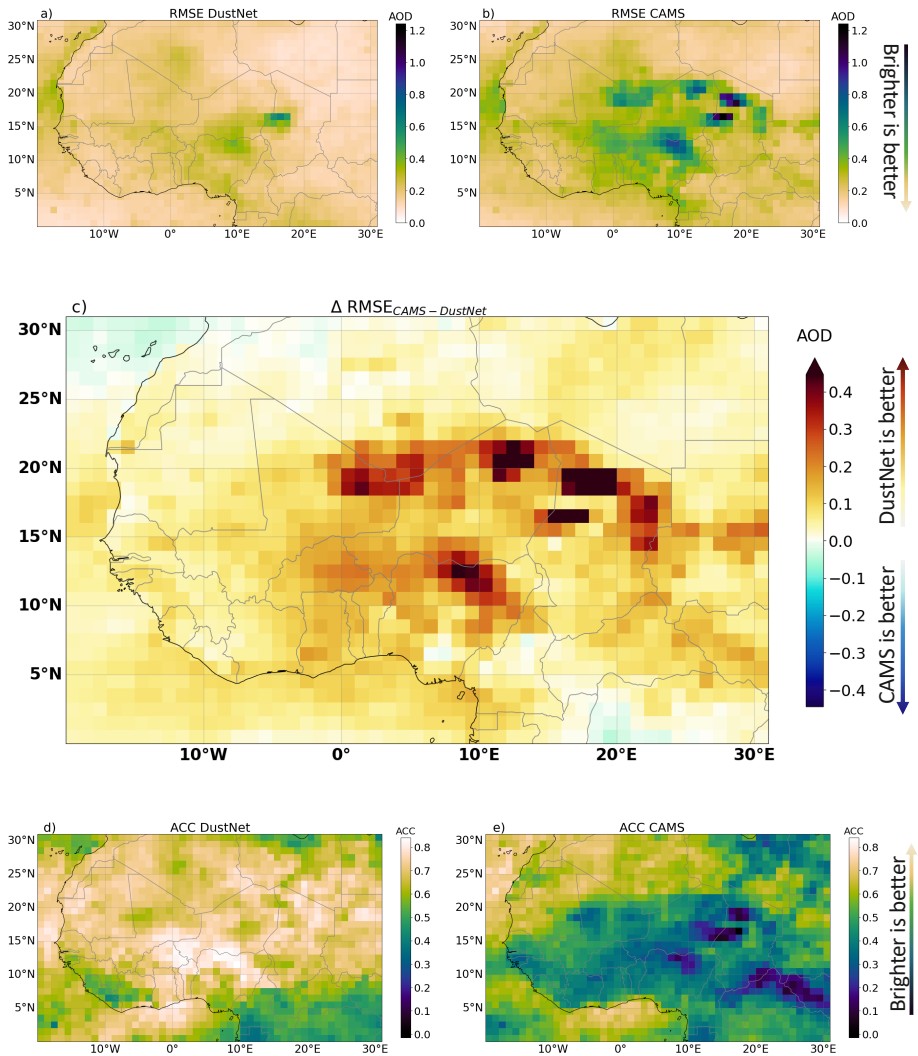

**Figure 3.** Metrics indicating model performance. Results for 24-hour (1-step) ahead predictions of daily AOD values (mean across the daily prediction time 2020-2022, n=1095) compared with the ground truth data from MODIS. The RMSE is shown for DustNet in **(a)** and CAMS in **(b)**, where the brighter the colour the smaller the error. Note, that the maximum error for DustNet is 0.62 AOD (medium green shades), while the maximum RMSE for CAMS reaches above 1.2 AOD (dark blue). In **(c)** the difference in RMSE between CAMS and DustNet where all yellow to deep brown shades indicate the advantage of DustNet, while the blue shades indicate the advantage of CAMS. White grid cells indicate locations where both of the models performed equally when compared to the ground truth data. Note the lack of deeper blue shades and the dominance of yellow and brown grid cells where DustNet outperformed CAMS. Lower panels **(d)** and **(e)** show the ACC for DustNet and CAMS respectively, where values above 0.6 (bright to white) indicate a valuable forecasting capability, while lower values (green to dark blue) indicate little to no predictive value. The ACC values in darkest blue shades indicate a misleading forecast.

We find that the DustNet model performs better in AOD forecasts than the physics-based CAMS model (Fig. 3). At nearly all spatial locations, DustNet predictions resulted in lower (better) RMSE values than CAMS during 2020-2022 (Fig. 3a and b). The greatest source of errors for both models was the most active dust source globally (Todd et al., 2007) — the Bodélé Depression (16.5°N, 16.5°E). Although this is the location of the highest error, here we show again that DustNet's RMSE is nearly 50% lower than that produced by CAMS (0.62 *versus* 1.24 respectively). The Bodélé Depression is of global importance for two main reasons: (i) it is responsible for over 50% of the dust generated from the Sahara desert (Todd et al., 2007; Washington et al., 2009; Jewell et al., 2021) and (ii) it was identified as the main source of minerals delivered seasonally to the Amazon basin (Koren et al., 2006; Jewell et al., 2021). A recent comparison of 14 physics-based models reveals their tendency to vastly underestimate the AOD forecast (ranging from -16% to -37%) in comparison to ground-based observations (Gliß et al., 2021). With nearly 40 million tonnes of dust emitted annually from the Bodélé Depression, lowering the forecasting error at this location, as achieved by DustNet, has the potential to vastly improve the forecasting of transported dust.

Overall, DustNet predictions outperformed CAMS forecasts on 95.26% of grid locations when comparing prediction errors (Fig. 3c). In Fig. 3c, grid cells in the darkest brown colour indicate locations where the errors produced by CAMS were over 0.45 AOD higher than that of DustNet, with the maximum error difference reaching 1.24 AOD. These locations represent central Saharan desert and arid regions, indicating the AOD composed of mineral dust, and thereby the more skillful ability of DustNet to capture dust generation. Moreover, DustNet captures the high mean AOD over northern Nigeria (associated with the seasonal Harmattan haze (Anuforom, 2007; Sunnu et al., 2008; Schwanghart and Schütt, 2008) more skillfully than CAMS (details in Section Performance of seasonal-mean forecast and Comparison of local predictions below). However, there are two locations at which CAMS forecasts performed better than DustNet (Fig. 3c). Both of these locations are adjacent to the boundaries (SE and NW corners), beyond which DustNet was unable to obtain information on the processes during training, while the data used to generate the CAMS forecast was extracted from a larger region (see Section CAMS forecast for details). Thus, the lack of information on processes at the boundaries may have affected the CAMS forecasts less than it affected DustNet. This, however, might be overcome by extending the study region for DustNet.

We also compare the ability of DustNet and CAMS to detect anomalies using the ACC, a quantitative metric used in previous similar studies (e.g. Lam et al., 2023; Bi et al., 2023). Here, DustNet also displays more skillful results than CAMS with a better (higher) ACC at 92.28% of grid cells shown in Fig. 3d and e. An ACC score above 60% is considered to be of value for forecasting purposes. The DustNet model surpasses this threshold at 79.89% of locations (white-yellow), indicating a better forecast value for a wider range of locations than CAMS (which had an ACC value above 60% at only 29.10% of the grid cells). Skillful detection of anomalies, combined with a high forecast value, indicates that the DustNet model could be a valuable addition to Earth System Models, where better representation of Saharan dust events leads to more realistic forecasts of precipitation and a better representation of the African monsoon (Anuforom, 2007; Düben et al., 2021; Balkanski et al., 2021).

Furthermore, we performed a comparative analysis of correlation coefficients between the forecasts and the ground-truth data. Figure 4 presents the daily correlation coefficients for two sets of comparisons: panel a) displays the correlation between MODIS-derived AOD values and DustNet predictions, while panel b) shows the correlation between MODIS and CAMS fore-

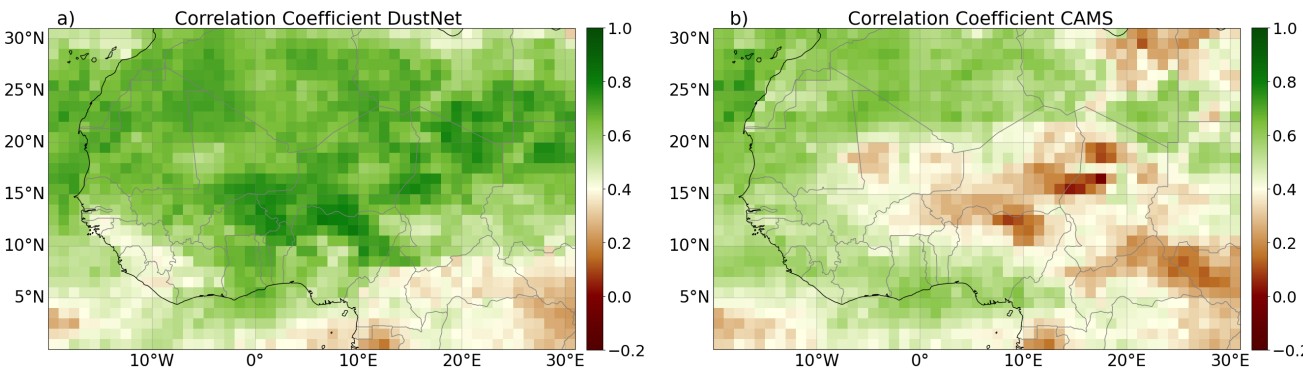

**Figure 4.** Daily correlation coefficients between MODIS AOD observations and model predictions are shown for **a)** DustNet and **b)** CAMS. The maximum correlation for DustNet is 0.82, with a minimum of 0.16, while the maximum correlation for CAMS is 0.75, with a minimum of -0.04. Values with weaker correlations ($\leq 0.4$) are represented in white to brown shades, whereas stronger correlations (>0.4) are depicted in green. The predominance of green shades, particularly over the Sahara region, highlights the advantage of DustNet predictions over CAMS

casts. Over the Saharan Desert, where mineral dust is the dominant contributor to AOD, DustNet exhibits a notably stronger correlation with MODIS, as indicated by the predominance of green shades. In contrast, CAMS demonstrates weaker correlations across the same region, evident in the presence of white to brown shades, which aligns with previously identified dust generation zones (highlighted in Supplementary Fig. S3). South of 8°N and east of 10°E, both models display weaker correlations. This outcome is expected, as DustNet's training data did not incorporate information on black carbon or secondary organic aerosols, which can seasonally influence AOD in the equatorial regions of Central African forests (Jo et al., 2023).

### 3.3 Performance of seasonal-mean forecast

Saharan dust aerosols are highly seasonal in emission and transport direction (Anuforom, 2007; Schwanghart and Schütt, 2008; Vandenbussche et al., 2020). Therefore, here we additionally compared the annual and seasonal means of DustNet predictions with MODIS and CAMS. Figure 5a shows the annual mean AOD values of MODIS and the model predictions. DustNet is capable of producing more realistic predictions in comparison to MODIS than the mean annual forecasts from CAMS. This is also confirmed by a highly significant correlation of the spatial mean AOD (DustNet: $r^2 = 0.91$; CAMS: $r^2 = 0.71$, in Appendix Fig. B1). The DustNet model also captures the high AOD generated from the dustiest spot on Earth, the Bodélé Depression, more precisely than CAMS in both annual and all seasonal means (darkest colours in all panels of Fig. 5).

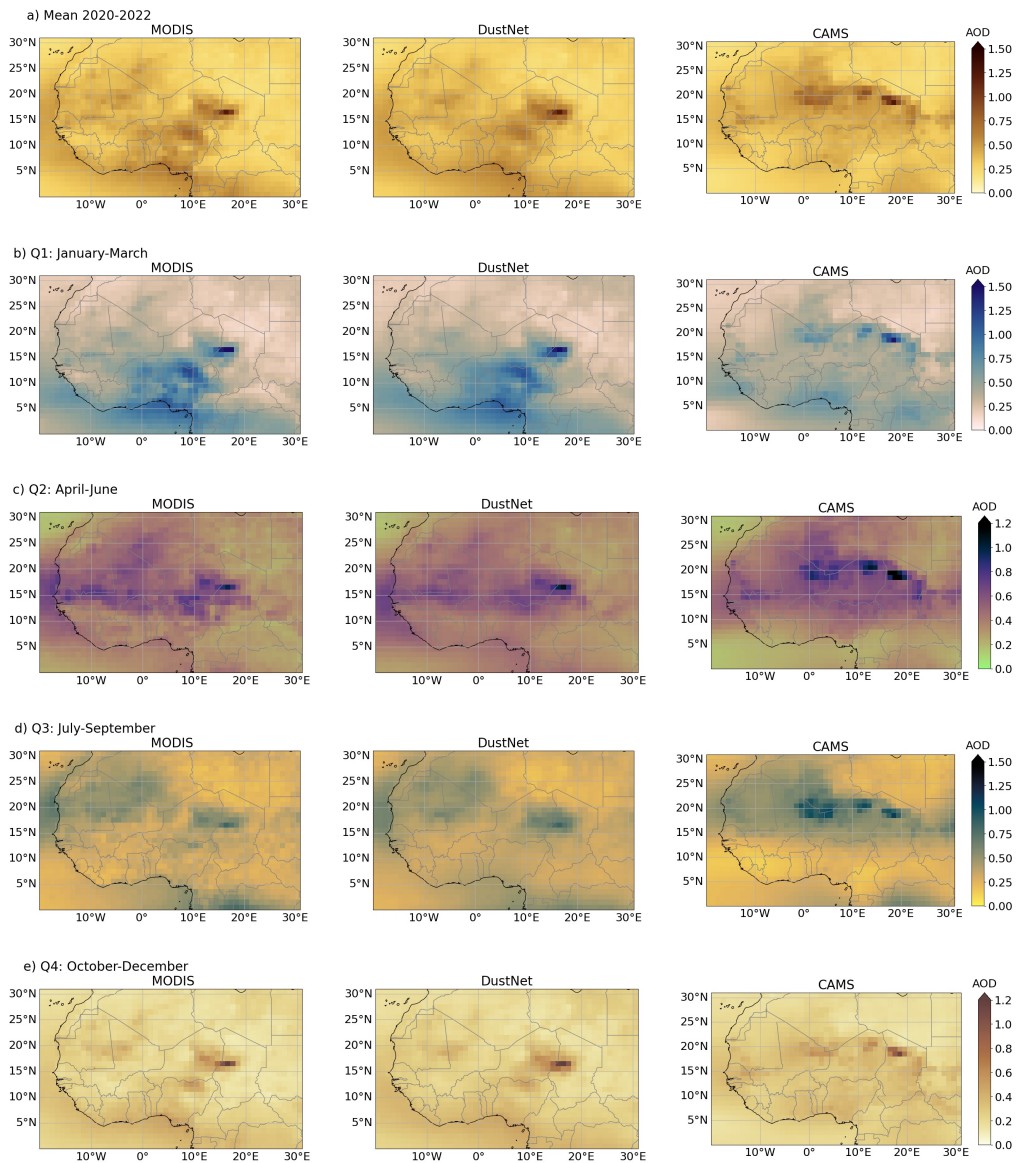

**Figure 5.** Annual and quarterly means of daily AOD values for 2020-2022. All mean AOD values were calculated from daily 24-hour ahead predictions. The **left** column represents AOD values from MODIS observations, predictions from DustNet are in the **middle**, while forecasts from CAMS are in the **right** column. **Row a)** compares the 3-year annual mean AOD between the observations and models. In **row b)**, the 3-year mean of daily AOD for Q1: January–March is shown, noting the main generation site of the Bodélé Depression (dark blue) and the southwestward transport of mineral dust. In **row c)**, these same means are shown but for Q2: April–June. **Row d)** shows that both models, CAMS and DustNet, skillfully detected the northward shift of mean AOD transport during Q3: July–September. In **row e)**, the seasonal decrease in aerosol activity for Q4: October–December is skillfully captured by both models when compared to observations from MODIS. Note here the change in the colour bar range.

In Fig 5, where the daily predictions were averaged to annual (a) and quarterly (b–d) means, we show that DustNet also captures the average seasonal displacement of AOD more skillfully than CAMS. During Q1 January–March (Fig. 5b), the influence of the Harmattan wind has a visible effect on the mean AOD with a south-westward transport of mineral dust from the main generation site of the Bodélé Depression (dark blue). Comparisons of AOD in Fig. 5b, c and d, indicate that DustNet captures this displacement more skillfully than CAMS. The seasonal shift of Saharan dust by $\approx 10°$ in latitude is consistent with past observations and studies (Prospero et al., 1981; Mbourou et al., 1997; Sunnu et al., 2008; Schepanski et al., 2017; Vandenbussche et al., 2020; Balkanski et al., 2021). Associated with a seasonal change in wind direction and large plumes of transported dust, this phenomenon is locally well known as the Harmattan haze and is responsible for the high increase in air pollution, especially around around Nigeria (Anuforom, 2007; Schwanghart and Schütt, 2008; Sunnu et al., 2008).

Previously noted mechanistic links between mineral dust and large-scale precipitation patterns, like the position of the Inter-tropical Convergence Zone (ITCZ) and the seasonal shift in the position of the West African monsoon, add to the importance of precise predictions of seasonal AOD displacement (Sunnu et al., 2008; Janicot et al., 2008; N'Datchoh et al., 2018; Balkanski et al., 2021). Additionally, seasonal means of the daily AOD, extracted from short forecast lead times of reanalysis models including CAMS, are used to validate other models including climate models (Zhao et al., 2022; O'Sullivan et al., 2020; Wu et al., 2020). Thus, achieving higher accuracy for the predictions of seasonal mean of daily AOD forecasts with DustNet could improve the performance of current forecasting models.

Long-term comprehensive comparisons (Gliß et al., 2021) show that the forecasts produced by physics-based models tend to underestimate the AOD values compared to MODIS ground truth observations. While this underestimation of AOD is clear between 5°N and 15°N, here we show that the CAMS forecast additionally tends to overestimate the AOD values around latitude 20°N over the Sahara during all the seasons of the period 2020-2022 (Fig. 5a-d, rightmost panel and Appendix Fig. B1b). This could be attributed to the locations of most of the ground observation stations, concentrated along latitude 10°N (Gliß et al., 2021).

The smoothness of predictions displayed by DustNet in comparison to CAMS is a characteristic of the regression algorithm used by deep learning models (explained in Bi et al., 2023).

### 3.4 Comparison of local predictions

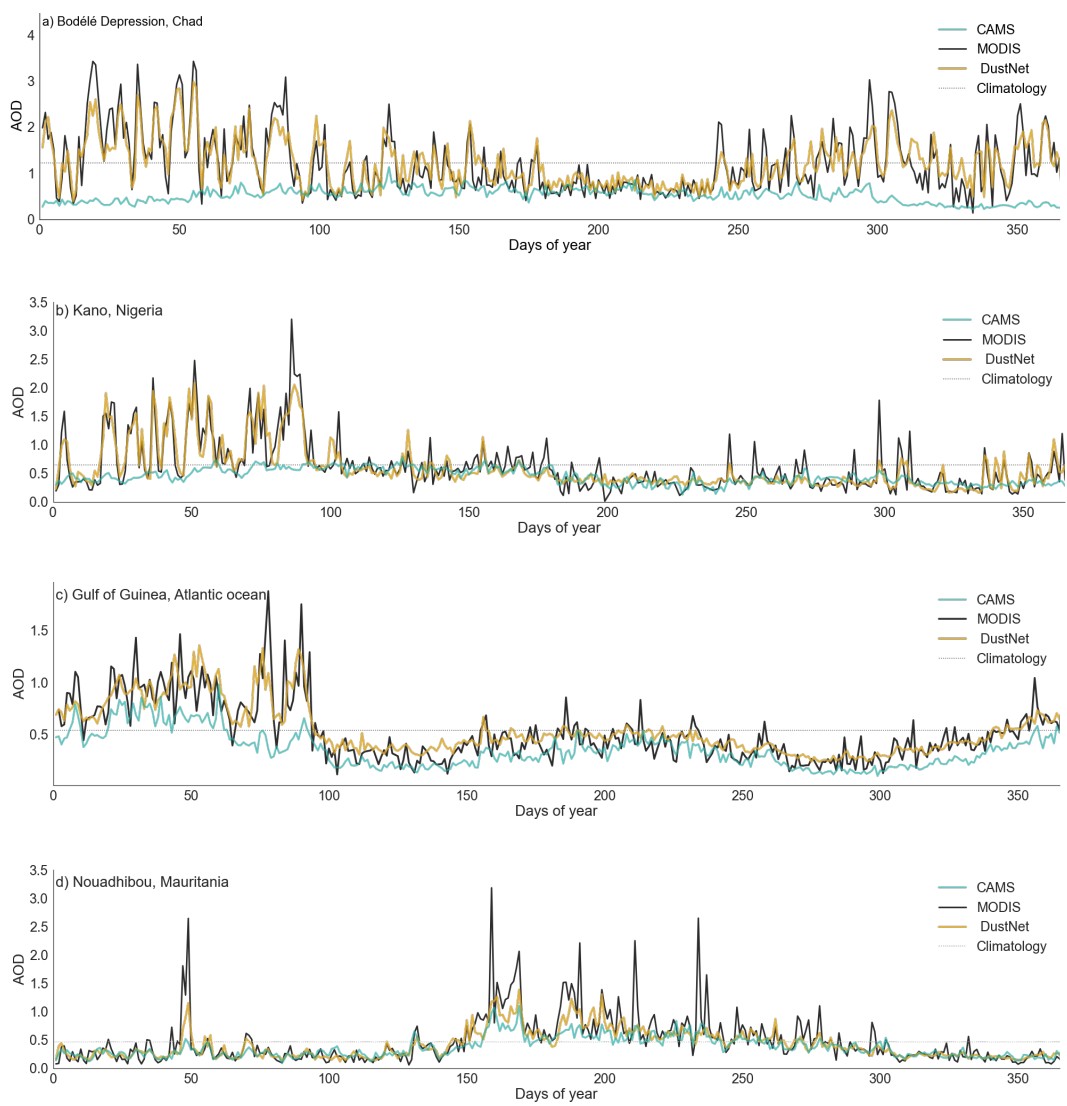

**Figure 6.** Local AOD predictions for each day of the year (2020-2022) for chosen point locations. Shown are daily means (2020-2022) of AOD predictions from DustNet (golden line) and CAMS (light-sea-green line) as compared to MODIS (black line) and climatological mean (dotted line). At all four locations predictions from DustNet are closer to MODIS values than CAMS forecasts. An increase in AOD can be seen in the first 90 days of the year in **a)** the Bodélé Depression, with lower but still elevated values towards **b)** Kano and **c)** Gulf of Guinea. These elevated AOD values during quarter 1 are not observed in **d)** Nouadhibou, which is consistent with the south-western direction of the Harmattan wind. DustNet also predicts daily and seasonal AOD variability at each site more skillfully than CAMS, whose forecasts tend to stay closer to or below the climatological mean. Both models struggle to fully capture the highest AOD peaks recorded by MODIS at the westmost location - Nouadhibou, however the DustNet model replicates these peaks better than CAMS.

We also test the ability of DustNet to provide accurate 24-hours (1-step) ahead predictions at four specific locations indicative
of the main dust transport routes (see Methods Section 2.6.5 for details on selected grids and locations). At all four locations,
DustNet predictions align with satellite data (MODIS) better than forecasts produced by CAMS (Fig. 6, and Appendix Fig. D1
for correlations). This is especially evident at the Bodélé Depression, despite the site producing the highest prediction errors
(see RMSE in Fig. 3a). The correlation between DustNet and MODIS at the Bodélé Depression is highly significant, with $r^2$ =
0.62, compared to CAMS which had $r^2$ = 0.01 (Fig. 6a and Appendix Fig. D1a). DustNet also skillfully detects the daily and
seasonal variability of the Bodélé Depression, demonstrating the ability of our model to skillfully capture dust generation at
this location. Similarly, 24-hour (1-step) ahead DustNet predictions for Kano, the second most populous city in Nigeria, align
better with MODIS ($r^2$ = 0.74) than forecasts from CAMS ($r^2$ = 0.12), whose predicted values stay close to the climatological
mean (Fig. 6b and Appendix Fig. D1b).

During the first quarter (Day Of Year 0∼ 90), the highest AOD values are present at the Bodélé Depression, Kano and the
Gulf of Guinea (Fig. 6c). In Kano, the AOD values are just slightly lower than at the Bodélé and slightly lower in the Gulf of
Guinea. Since both Kano and the Gulf of Guinea are positioned south-west from the Bodélé, their corresponding AOD values
during quarter 1 indicate the Bodélé Depression as a generation source (Schepanski et al., 2007; Jewell et al., 2021; Kok et al.,
2021a). This also shows the ability of DustNet to capture generation and transport of AOD consistent with shifts in seasonal
wind direction indicated in past studies (Schepanski et al., 2017; Schwanghart and Schütt, 2008; Anuforom, 2007; Sunnu
et al., 2008). During the third quarter (DOY 180∼ 270), however, DustNet struggles to correctly capture the highest peaks in
Kano and the Gulf of Guinea. The seasonal shift in meteorology and especially wind direction at these locations leads to an
AOD composed of a mixture of aerosols, including sea-salt, and black carbon from biomass burning and industrial pollution
(Anuforom, 2007; Mari et al., 2008; Knippertz et al., 2017). An area of future research could include information on vegetation
and land cover during the training process, which would allow the model to distinguish between the ocean, Sahara Desert
and central African forests. This would likely improve predictions for these regions and other aerosol species in general. The
highest AOD values are also missed in Nouadhibou (Fig. 6d) during quarter 3 (DOY 180∼260). However, here the seasonal
increase in AOD points to a more localised origin, since dust generation at the Bodélé Depression is at its lowest with a daily
AOD ≤1.0. This finding is consistent with past analyses of boreal summertime dust generation, which point towards Western
Sahara, Mauritania, Algeria and Mali as dust sources (Schepanski et al., 2007; Friese et al., 2017; Jewell et al., 2021; Kok et al.,
2021a).

### 3.5 Feature importance

Assessment of feature importance, shown in Fig. 7, reveals that the 1-day lag of AOD emerges as the single most important
feature, as removing it leads to the largest increase in MSE (0.00343), emphasizing the DustNet's strong reliance on recent
AOD state. Vertical velocity at 850 hPa follows closely (MSE 0.00246), underscoring the role of mid-level atmospheric motion
in controlling aerosol transport. Other prominent features include the v-component of wind at 850 hPa and wind speed at 1000
hPa, illustrating that both near-surface and lower-tropospheric winds are vital for accurate model prediction. Additionally, the
significance of vertical velocity at 550 hPa and wind power at 1000 hPa highlights how stronger vertical movements and more

energetic surface-level flows further amplify AOD generation processes. Finally, a 2-day lag of AOD only comes seventh in our feature importance, suggesting that while longer AOD histories still add predictive value, the model prioritises more immediate conditions.

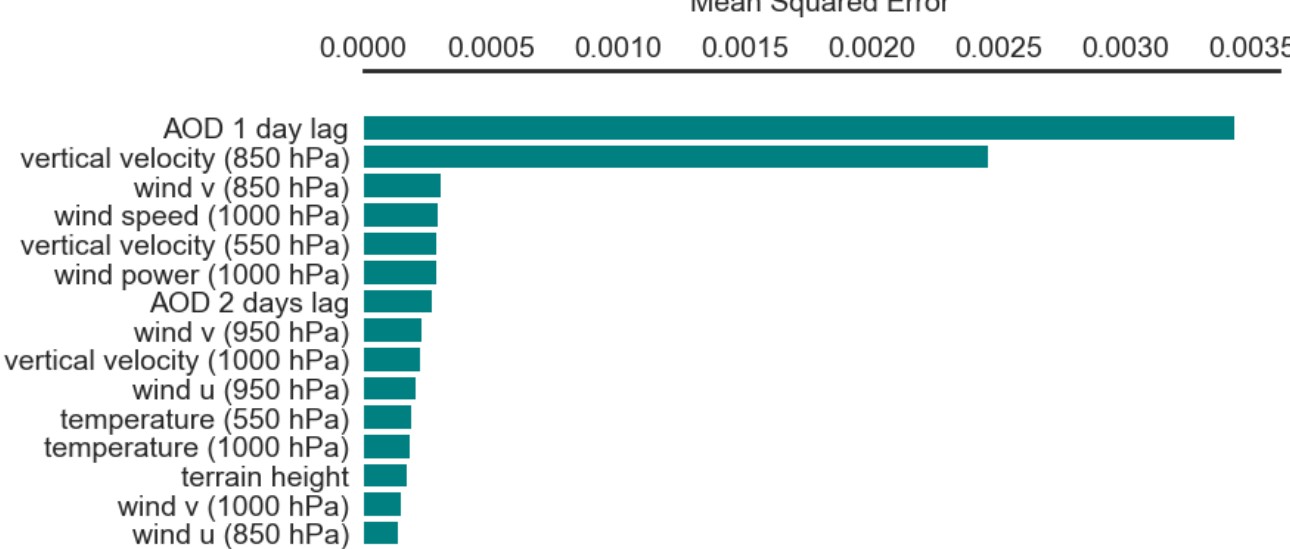

**Figure 7.** Results of the feature importance analysis for the DustNet model, based on mean squared error (MSE), highlighting the 15 most influential input features. Features yielding the highest MSE when removed (zeroed-out) are deemed most critical for the DustNet model's 24-hour (1-step) ahead predictions. The bar chart shows that omitting the 'AOD 1-day lag' feature leads to the highest increase in MSE, followed closely by 'vertical velocity at 850 hPa'. These findings indicate that the input channels associated with the recent AOD information and mid-level atmospheric motion affect DustNet's forecast accuracy, underscoring their importance in predicting daily AOD over the region.

The next most critical features among the top 15 include temperature at both 550 and 1000 hPa, terrain height, and the year sine/cosine signals. Notably, temperature at these two pressure levels is more influential than at intermediate levels, implying that near-surface heat fluxes and upper-level thermal profiles strongly affect the DustNet model predictions. Meanwhile, the prominence of terrain height underscores the importance of local topography for channelling orographic flows or indicating primary aerosol source regions – an effect that DustNet treats as more significant than AOD beyond two days in the past. Similarly, the inclusion of seasonal features (year sine and cosine) among the top 15 indicates that periodic patterns, contribute considerably to AOD activity in this region. Together, these findings reveal that immediate AOD conditions, vertical motions, surface-level wind intensity, and broader seasonal cycles collectively govern short-term AOD forecasts of our DustNet model.

In contrast, the remaining features, such as relative humidity and wind power at non-surface levels, show noticeably less influence on day-to-day predictions (see Supplementary Fig. S5). Their lower importance indicate an overlap with the dominant factors like vertical velocity or near-surface wind fields, which effectively capture much of the variability in predicted AOD.

Nonetheless, interpretation of feature ranking results requires caution, as perturbation-based methods evaluate each input independently and may overlook intricate interactions among correlated features. Consequently, certain features might appear less important if their effects are partially hidden by the stronger predictors. Overall, these results confirm that recent AOD states, low to mid level atmospheric dynamics, terrain height and seasonal signals form the principal pillars of DustNet's predictive skill for 24-hours ahead AOD.

## 4 Discussion and future developments

The fast and skillful short-term predictions with DustNet present an opportunity for the forecasting community to incorporate a comprehensive aerosol scheme into future forecasts. The current coarse representation allows for quick testing and replication by professionals and enthusiasts alike. DustNet also skillfully captures aspects of atmospheric processes such as dust generation, transport, or seasonal variations, when compared to the satellite data. Furthermore, skillful representation of atmospheric aerosols at specific locations opens a possibility for DustNet integration into more localised weather models.

The specific DustNet model architecture may be used for predicting other atmospheric particles, or indeed other environmental phenomena. However, this would require retraining the model using input features that represent the chosen particle or phenomenon. For example, to capture aerosols due to black carbon, features such as land cover types, vegetation, leaf area index, and forest fire locations should be considered. Similarly, when aiming to capture atmospheric aerosols due to sea-salt particles, features including wave height, energy flux into waves, peak wave period, and ocean surface stress should be taken into account. Moreover, DustNet model architecture may be used for predicting other spatio-temporal dynamics, such as phytoplankton concentrations from satellite-derived chlorophyll-a data, by substituting input variables with relevant meteorological and ocean state data.

While DustNet outperforms CAMS in short-term forecasts, it is not without limitations. Although the model is trained on 43 features, only one - terrain - represented the ground conditions. Thus, incorporating additional information could be beneficial in capturing more nuanced or indeed wider interactions. For example, the generation of dust depends not only on the atmospheric conditions, but also on the soil moisture, soil type and mineral composition from which atmospheric dust derives (Knippertz et al., 2017; Van Der Does et al., 2018). Soil type and mineralogy impact the dust interactions with other atmospheric particles, and wider Earth systems by delivering essential minerals to the oceans and rainforests (Kok et al., 2023; Jickells et al., 2014; Koren et al., 2006). Information on ground vegetation and cover can also play a role in determining dust generation locations and transport, especially over forests and in urban areas.

Additionally, DustNet's predictions at the northern and southeastern locations of the region boundaries are visibly weaker than at the centre (Fig. 3c and 3d). The predominant wind and transport directions of the atmospheric dust during this study are confirmed as west and southwest (Fig. 5, especially 5b and 5c), which indicates that the northern and southeastern areas may be governed by processes not included in the feature selection of this study. This is not surprising, since the Mediterranean Sea is directly to the north of our study region, while the Congolian rainforest covers grids directly to the south and southwest of the

boundaries. These indicate the potential for more skillful forecasts with a broader study area, which, together with additional features, could capture more nuanced processes above the oceans and rainforests.

Likewise, the daily predictions of extreme AOD values at point locations (especially in Nouadhibu, Fig. 6d) can fall short of the values captured by the satellites. Together with the deterministic nature of the model, DustNet's predictions lack the probability distribution with the length of the tail for the extreme values.

Addressing these limitations is crucial for future advancements. Rather than increasing the model's training time or epochs, we propose expanding the training data with diverse geographical information. This approach would capture nuanced interactions of atmospheric dust with Earth's systems. The inclusion of data sources from broader environmental disciplines, expanding study locations, and extending lead-time predictions are important next steps. Thus, a multidisciplinary approach can further enhance DustNet capabilities and contribute to a range of specialised AI models with skillful predictions.

## 5   Conclusions

This study introduces a novel application of neural networks to improve the prediction of aerosols over the Saharan Desert, the world's most significant source of atmospheric dust. Dust aerosols play a critical role in global climate systems, air quality, and ecosystems, yet traditional models often struggle with accuracy and speed due to the complex nature of dust dynamics and computational burden.

The research employs machine learning to bridge these gaps, offering a method that is both efficient and accurate. By training the DustNet model on satellite-based and reanalysis datasets, the research demonstrates significant improvements in capturing spatial variability of dust emissions. The results show that the neural network can produce skillful predictions while requiring fewer computational resources than conventional models.

Moreover, the framework is designed for accessibility and reproducibility, utilising open-source tools and emphasising transparency to facilitate broader adoption within the scientific community. This work not only advances the predictive capabilities for dust aerosols but also serves as a template for applying machine learning to other challenging atmospheric problems. Its potential implications span across atmospheric research, and practical applications like air quality management.

*Code and data availability.*  The full Python code for each model (DustNet, U-NET and Conv2D) with structured input data Nowak et al. (2024a) were deposited in Zenodo and are publically available at https://zenodo.org/records/10722953. The repository includes all results from the DustNet model (output data), and Jupyter Notebooks with Python code to replicate all statistical analysis to reproduce each figure included in this article. Pre-processed ERA5 and AOD data (Nowak et al., 2024b) are deposited as NumPy files in Zenodo together with Python imputation code at https://zenodo.org/records/10593152.

Reanalysis of atmospheric features were downloaded from the Copernicus Climate Data Store collection 'ERA5 hourly data on pressure levels from 1940 to present'. Unprocessed datasets are available from Copernicus Climate Change Services (C3S) Climate Data Store (CDS) at https://cds.climate.copernicus.eu/cdsapp/. Pre-processed ERA5 data is also included in the aforementioned Zenodo repository.

The AOD at 550nm Level 3 daily data for combined Dark Target and Deep Blue algorithms were retrieved from Moderate Resolution
        Imaging Spectroradiometer (MODIS) on both Aqua and Terra spacecraft. Both datasets are available from NASA's Atmosphere Archive &
        Distribution System (LAADS) Distributed Active Archive Center (DAAC). Both MOD08_D3 and MYD08_D3 files can be retrieved from
        https://ladsweb.modaps.eosdis.nasa.gov/search/. Pre-processed AOD data is also included in the aforementioned Zenodo repository.
        The forecast of AOD was downloaded from the Atmosphere Data Store of Copernicus Atmosphere Monitoring Service (CAMS). The total
aerosol optical depth at 550nm from the Global atmospheric composition forecast for midday run with a 24-hour lead-time can be obtained
        from https://ads.atmosphere.copernicus.eu/#!/home.

## Appendix A:  CNN models schematic

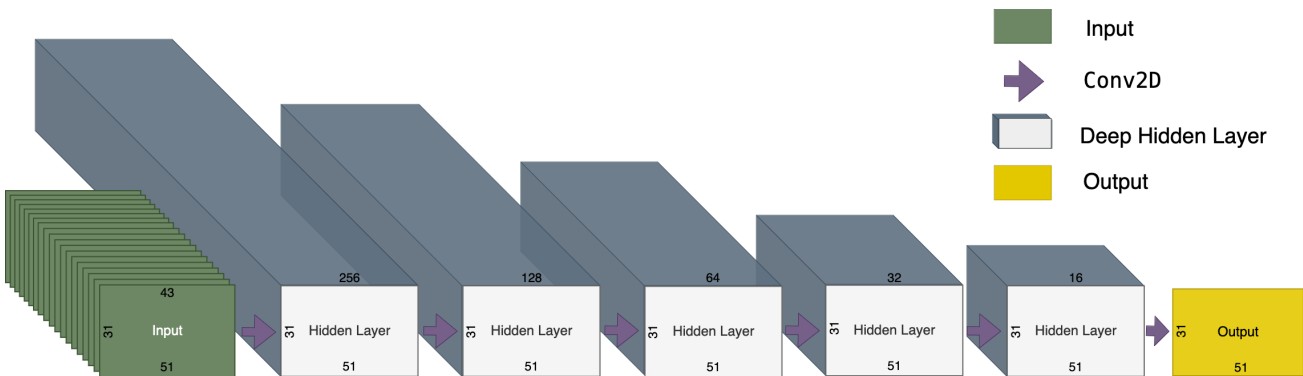

**Figure A1.** Schematic representation of simple Conv2D model. **From left:** the Input Layer of shape (31,51,43) is represented in green.
Following, are the 5 Hidden Layers of this same width and height as Inputs, but with different depths. The depths (number of hidden
connections) were set in decreasing order to 256, 128, 64, 32 and 16. The last 2D Convolution with depth 1 created the output which shape
was matching our target AOD.

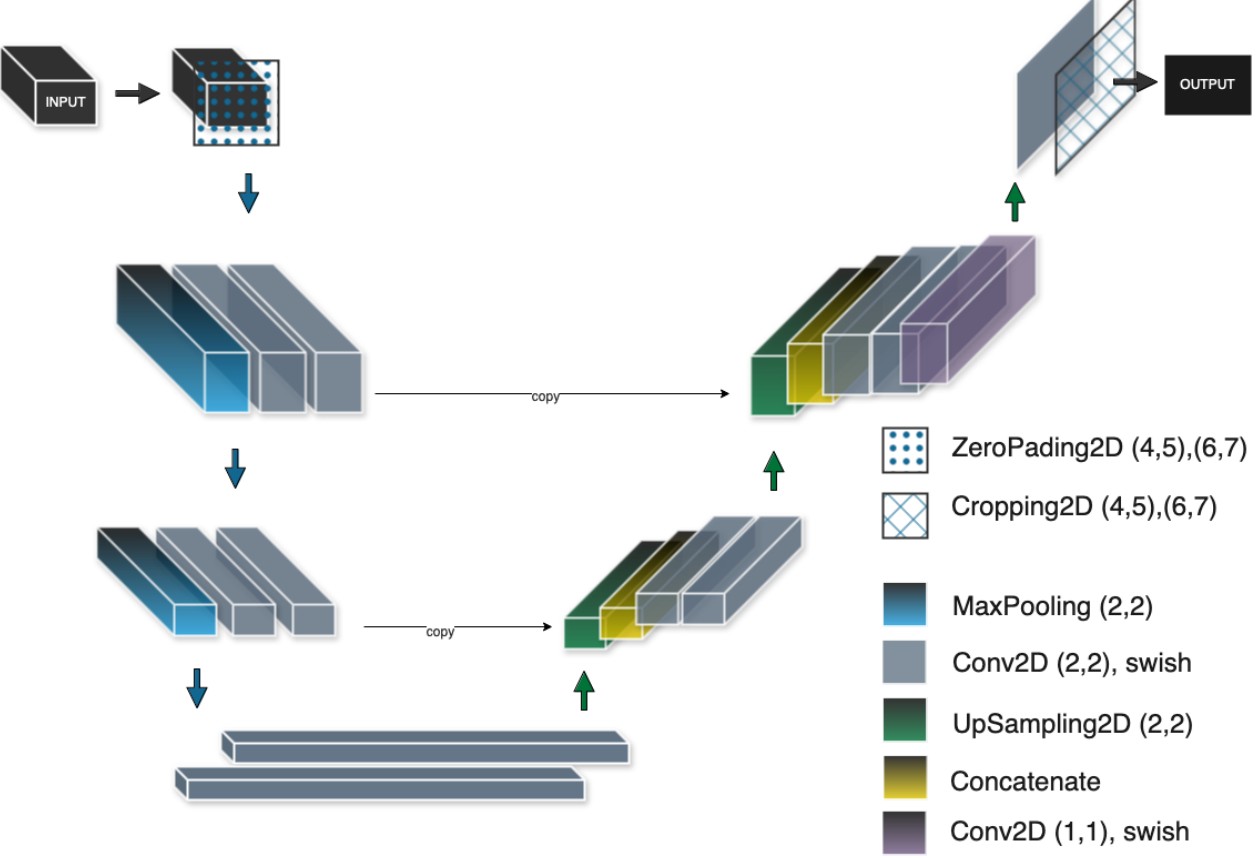

**Figure A2.** Illustrative sketch of U-NET model architecture with individual blocks representing model layers. The input layer (31,51,43) is first padded with 2D zeros layer which increases the height and width of input shape (40,64,43). The encoding pathway (blue arrows down) includes 2 successive layers of Conv2D which increase the depth of the input size $(40 \times 64 \times 64)$. Following MaxPooling layer decreases the first two dimensions, while Conv2D increases the third $(20 \times 32 \times 128)$. After the second MaxPooling and double Conv2D the input is reshaped to $(10 \times 128)$. The decoding pathway (green arrows up) includes 2D Upsampling and Concatenation which now increases the width, height and depth to $(20 \times 32 \times 384)$. Following 2 layers of Conv2D decrease the depth while UpSampling and Concatenation increases the shape to $(40 \times 64 \times 192)$. The last two layers of Conv2D decreased the depth to $(40, 64, 64)$ while its final layer brought the depth down to $(40 \times 64 \times 1)$. Last layer, Cropping2D ensured the output matched the target size of $(31 \times 51 \times 1)$.

## Appendix B: Temporal analysis

When the data were spatially averaged over the study area for each test day, both DustNet and CAMS revealed high correlation with MODIS observations. However, DustNet's predictions exhibited stronger correlation with MODIS observations, achieving $r^2 = 0.91$ (see Fig. B1a) in comparison to CAMS. This high correlation indicates that DustNet effectively captures the daily variability of AOD across the Sahara, however with slight tendency to overestimate the high AOD values. In contrast, CAMS forecasts, while still highly correlated with MODIS ($r^2 = 0.71$), display a more frequent tendency to underestimate both low and high AOD values (Fig. B1b), and overestimate middle AOD values more frequently than DustNet. Both model results were highly significant, with $p-value \leq 0.0001$.

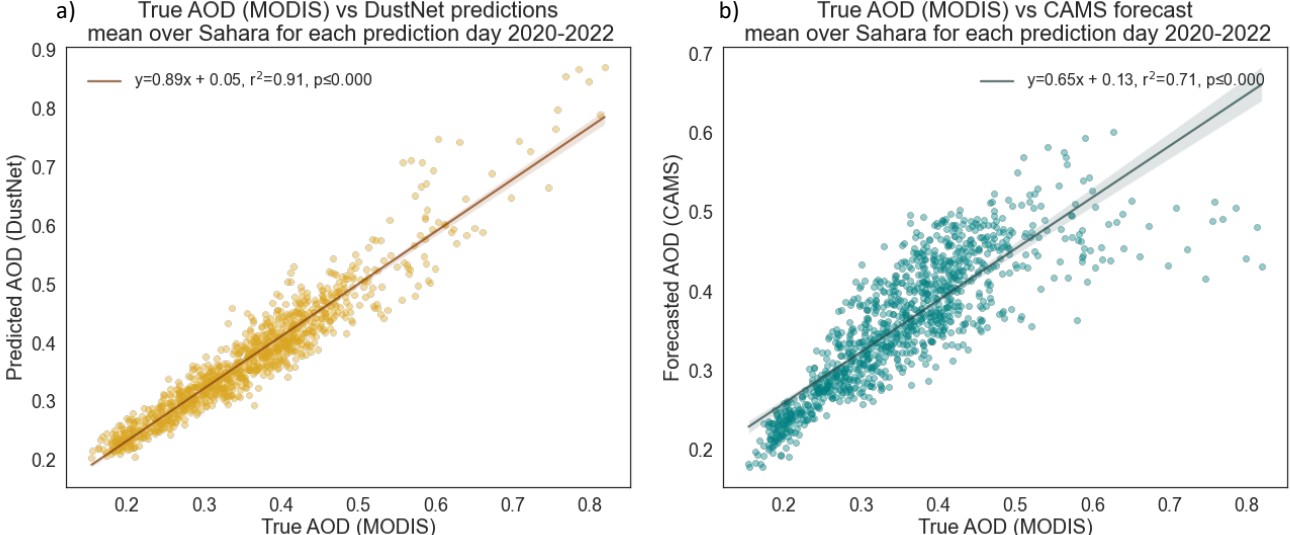

**Figure B1.** Spatially averaged daily AOD (2020-2022, n=1095) regressed between model predictions and MODIS data. Linear regression with corresponding y equation, Pearson's $r^2$ and p values were calculated for daily spatial mean AOD over the Sahara for 2020 - 2022. Shown in **a)** AOD prediction results from DustNet correspond to MODIS data well with high $r^2 = 0.91$, and only a slight tendency to overestimate higher AOD. In **b)** the mean AOD forecasts from CAMS are shown to correspond with MODIS data well, $r^2 = 0.71$ though, with more frequent tendency to underestimate both low and high AOD values. Results from both predictions are highly significant with p< 0.0001.

Figure B2 shows the comparison of mean RMSE and mean bias errors (MBE) which further underscores the advantage of DustNet predictions over CAMS. At all time steps, DustNet consistently achieves lower RMSE values than CAMS, reflecting its improved predictive accuracy. Moreover, the MBE of DustNet fluctuates closer to zero, indicating a lower systematic bias compared to CAMS, which tends to deviate more frequently from the true AOD values. Together, these results confirm that DustNet provides more skillful deterministic AOD forecasts, both in terms of overall accuracy and reduced bias.

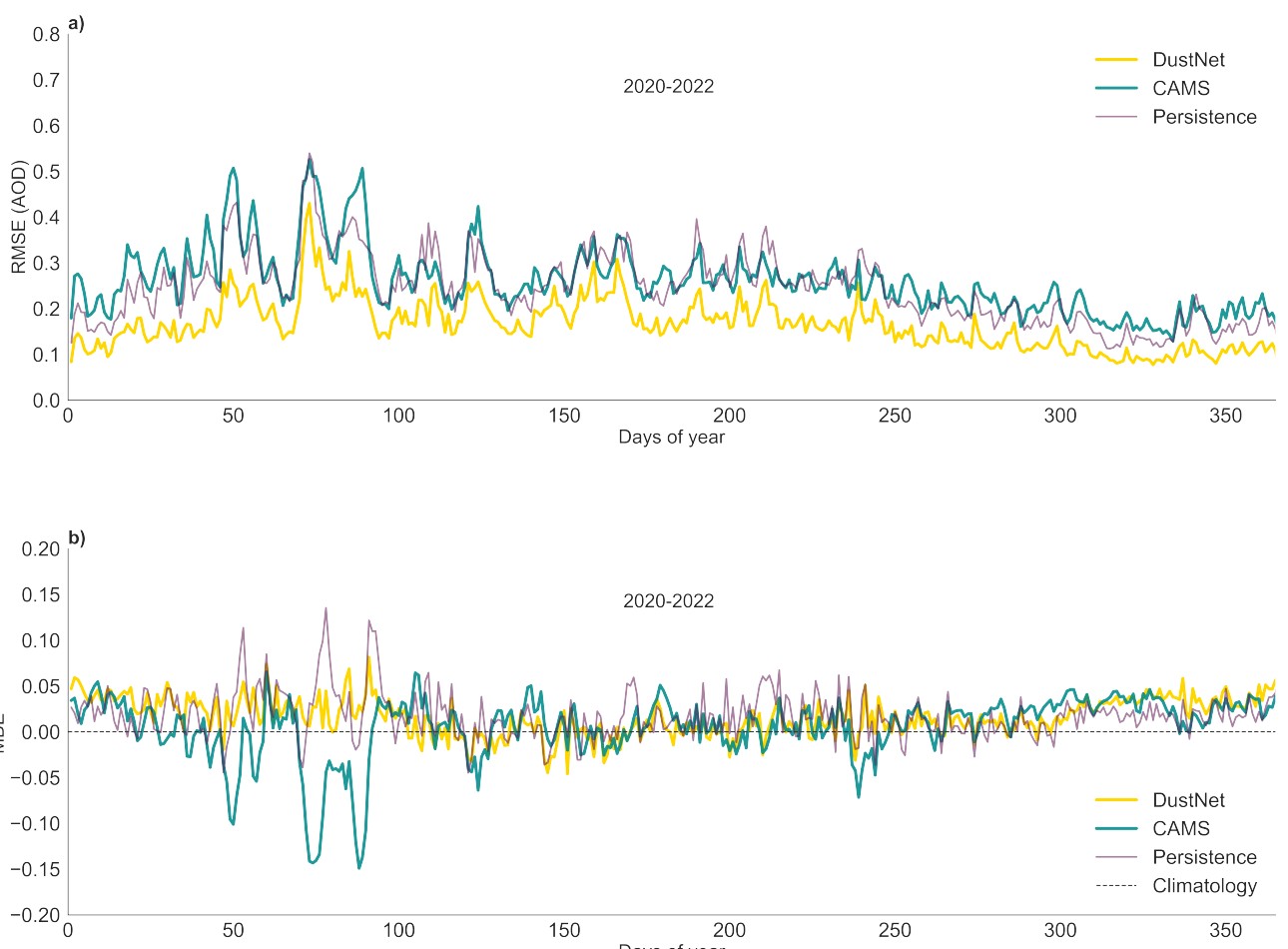

**Figure B2.** The top panel **a)** presents the study area mean RMSE calculated from daily AOD values predicted by DustNet (yellow), CAMS (cyan) and corresponding persistence (plum) are shown. At all time-steps the DustNet model predictions show smaller (better) errors than those produced by CAMS and persistence. The lower panel **b)** shows the temporal mean bias errors (MBE) from the DustNet predictions (yellow), CAMS (cyan) and persistence (plum). Here, the DustNet bias fluctuates close to zero more often than the bias produced by CAMS and persistence.

## Appendix C: Spatial analysis

Results presented in Fig. C1 indicate that DustNet predictions systematically show lower bias (lighter shade) than CAMS forecasts.

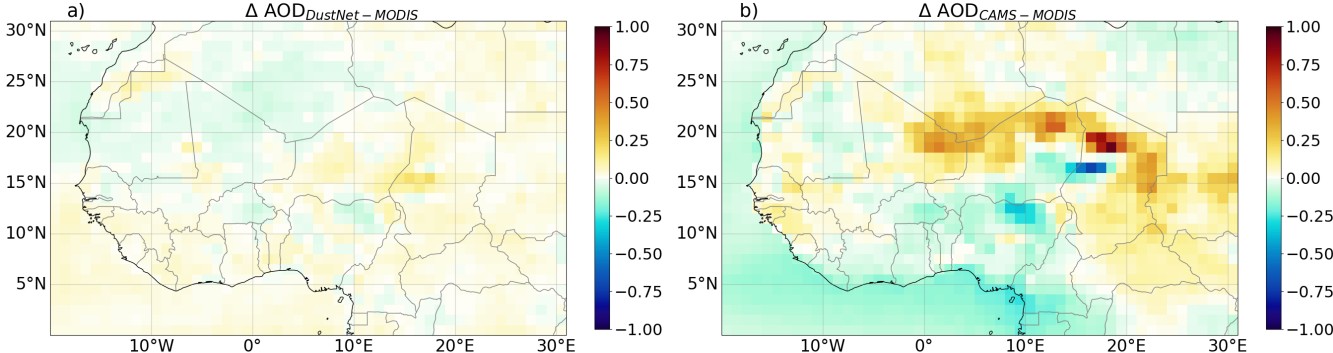

**Figure C1.** Bias of daily predictions for **a)** DustNet and **b)** for CAMS with respect to MODIS data (n=1095). The lighter the shade, the lower the bias. Note that the maximum bias produced by DustNet is 0.21 while the maximum bias for CAMS is 0.93. The areas of AOD over-prediction in comparison to MODIS are shaded in yellow-brown, while under-predicted AOD is shaded in blue.

## Appendix D: Local predictions - daily temporal analysis

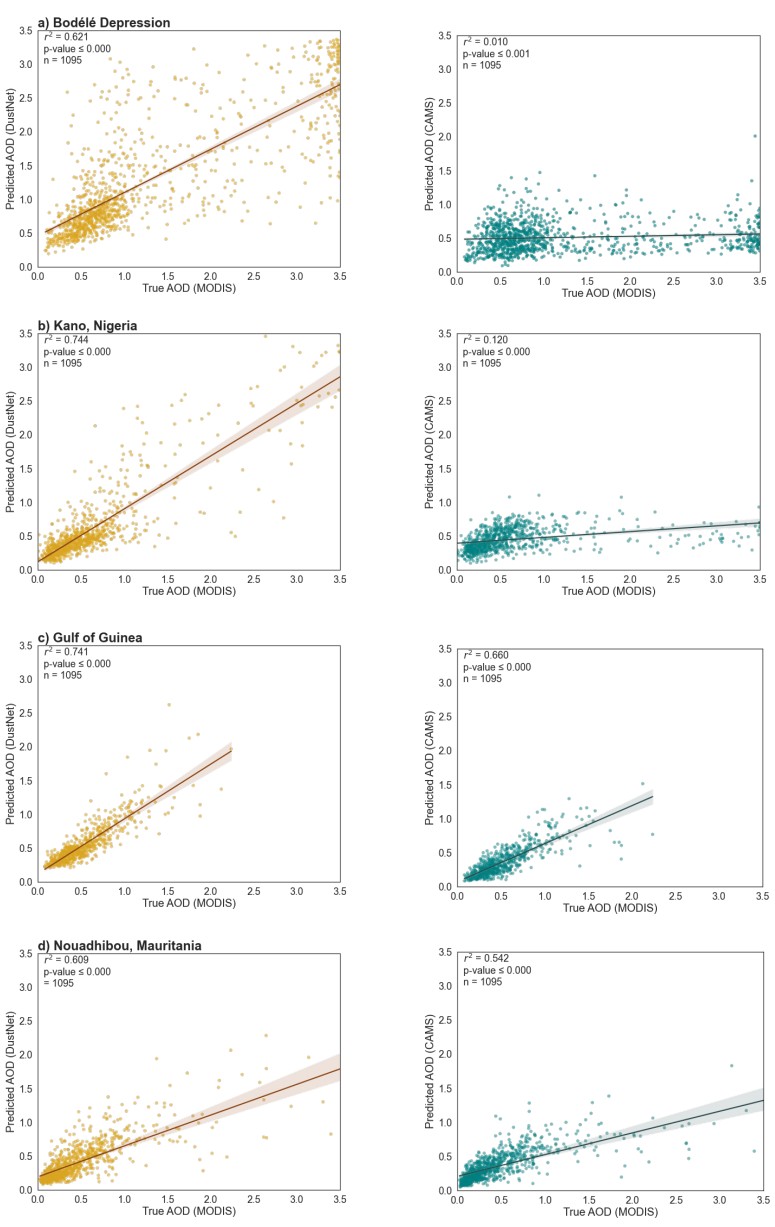

**Figure D1.** Scatter plot relationship between predicted mean AOD values (2020-2022) and MODIS data at four selected locations. Results for DustNet (**left panel**) and forecasts from CAMS (**right panel**) at all four locations show better agreement of DustNet predictions with MODIS data. In **a)** the Bodélé Depression, Chad - highest source of dust in the Sahara, here DustNet is significantly better than CAMS; **b)** Kano, Nigeria – the second most populous province **c)** Gulf of Guinea – over the ocean; and **d)** Nouadhibou, Mauritania – coastal location.

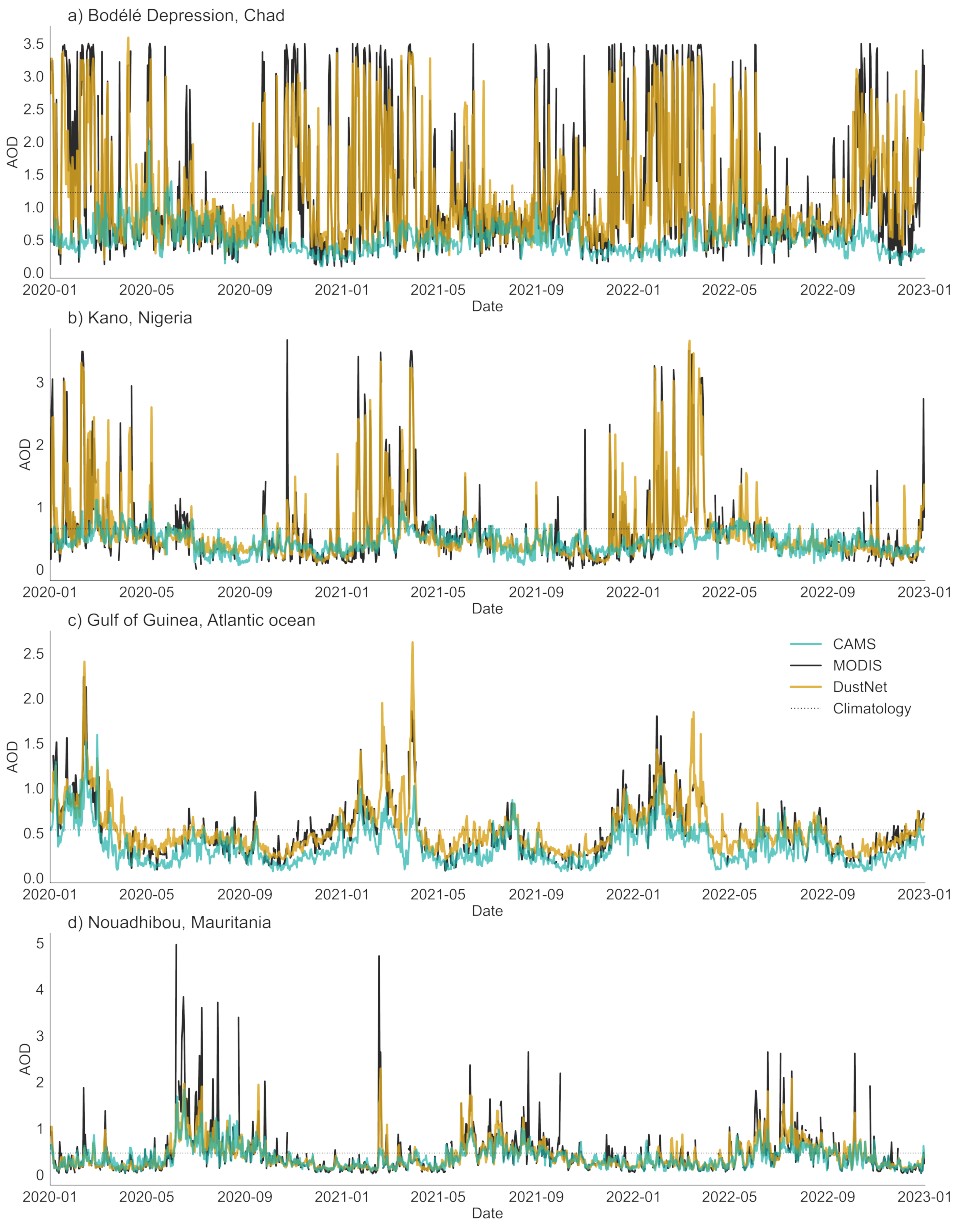

**Figure D2.** Same as Fig. 6, but for daily data at each selected location. Note that the AOD at the Bodélé Depression (panel a) reaches a hard maximum of 3.5 – an artifact of the Level-3 MODIS retrieval algorithm, which caps values beyond this threshold. Consequently, DustNet predictions also never exceed 3.5 AOD at this location.

*Author contributions.* Conceptualization: T.E.N., S.S., B.I.S., A.T.A. Methodology: T.E.N., S.S. Investigation: T.E.N. Visualization: T.E.N. Supervision: S.S., B.I.S., A.T.A. Writing—original draft: T.E.N. Writing—review and editing: T.E.N., S.S., B.I.S., A.T.A.

*Competing interests.* The authors declare no competing interests.

*Acknowledgements.* This work was supported by the UKRI Centre for Doctoral Training in Environmental Intelligence, Engineering and Physical Sciences Research Council Grant Reference: EP/S022074/1. We acknowledge NASA for producing, maintaining and releasing the MODIS AOD data which was used for training and comparison in this study. For these same reasons we acknowledge Copernicus Atmospheric Monitoring Service and ECMWF for their open release of CAMS AOD data. We would also like to acknowledge the reviewers of this paper: the Anonymous Referee #1 and Prof Narendra Ojha – Referee #2, whose comments contributed to better communication of our results and an overall improvement of this paper.

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
