# Peer review of "DustNet (v1): Skillful neural network predictions of dust aerosols over the Saharan Desert"

_EGUsphere, 2024_

## Author Comment (AC2)

Feature importance based on MSE

---

## Author Response (AR1)

Wednesday 25 December 2024

Skillful neural network predictions of Saharan dust
Trish Ewa Nowak, Andy T. Augousti, Benno I. Simmons, and Stefan Siegert
Pre-print: https://doi.org/10.48550/arXiv.2406.11754

Dear Editorial Team,

We would like to thank both referees for their thorough evaluations and detailed, constructive comments. These will inadvertently improve the quality of our manuscript.

Below, we include the original referees' comments in black font and our point-by-point responses in dark cyan. In addition to the suggested changes, we have made some minor adjustments to the figure notations, replacing capital letters with lowercase, and added section numbers. For clarity and ease of reference, we have also included line numbers (or pages) in amethyst font, which indicate additions, removals, or replacements made to the original text. Please note that the line numbers correspond to the marked-up version of our revised manuscript.

Best wishes,

**Trish Nowak**
*PhD Student*
*UKRI CDT in Environmental Intelligence: Data Science & AI for Sustainable Futures*
*University of Exeter*
*https://mathematics.exeter.ac.uk/staff/pn284*

**Referee #1**

RC1: 'Comment on egusphere-2024-2259', Anonymous Referee #1, 26 Sep 2024

The authors present a neural network model that predicts MODIS AOD retrievals over the Sahara based on ERA5 meteorology. The topic is relevant and my overall impression is that the model performs well in its task and the study is complemented with well documented code and data. However, there are several aspects that should be discussed or clarified before publication.

**Major comments:**

1. Title

Contrary to what the title and model name suggest (and to related studies, e.g. [1][2]), the model does not use any dust-specific data or model architectures, apart from the choice of a domain strongly affected by Saharan dust. Furthermore, the study area is not only affected by desert dust, but also by biomass burning aerosol. Therefore, the model presented here is more of an AOD model demonstrated over the Sahara. I suggest to adjust the title and the introduction of the manuscript accordingly.

This is an excellent point, and we thank the reviewer for raising it. It is indeed true that the model uses aerosol optical depth as a proxy for dust over the Saharan Desert during training. Thus, we have reflected this in both, the manuscript introduction and the title to read:

Lines 1-2: 'Skillful neural network predictions of dust aerosols over the Saharan Desert.'

**2. Conv2D and U-NET**

The model descriptions (pages 14 and 15) are not detailed enough, especially for the alternative models (Conv2D and U-Net). The exact model architectures and hyperparameters are not clear and providing schematics of the Conv2D and U-Net models would be very helpful in comparing the three models. All details are available in the code and data repository, but the model architectures should be clear from the manuscript itself.

We are grateful to the reviewer for bringing this point to our attention.

We agree that the manuscript would benefit from additional schematic drawings of the models used.

Pages 39-40: 'Therefore, we have added Figures 15 and 16 (A11 and A12), which include schematic drawings of the Conv2D and U-NET model architectures, to the Supplementary Materials.

**3. ZeroPadding & Kernel**

Some model design choices appear questionable. Zero padding of the input essentially introduces artificial, unrealistic input values at the boundaries, so that the approach is no longer translationally invariant, limiting the benefits of convolutional models. In addition, the 2 x 2 kernels collect data asymmetrically around each grid cell. For example, in the case of the Conv2D model, the AOD prediction for each grid cell appears to consider input from only one quadrant of its surroundings. Please discuss whether these are problems or why they are not. Not problematic, but apparently not necessary, is the imputation to the AOD observations. Instead, missing values could be excluded from the loss calculation.

We thank the reviewer for raising these interesting questions.

**Re: ZeroPadding2D Layer**
This layer extends the image size at the borders by adding artificial zeros, allowing the kernel (the convolving window) to fully consider the grid cells at the original border of the input image. By enabling the convolving window to move beyond the original image size, the predicted image becomes larger than the input image. This larger output is then readjusted after the final convolution layer using Cropping2D. Thus, the ZeroPadding2D layer improves predictions at the image edges.

We invite the reviewers to refer to supplementary Figure 7 (A3), which illustrates the bias in DustNet model predictions and CAMS forecasts. As shown, the ZeroPadding2D layer does not lead to noticeable biases along the edges.

**Re: 2x2 Kernel**
Using a kernel size with a divisible stride in Conv2D produces smoother image predictions without the checkerboard effect [4]. Refer to Figure 12.21 on page 404 of [4] for details. In our unique use of Conv2D and Conv2D Transposed, we employed a kernel size of 2x2 with a stride of 1x1. Testing with larger kernel sizes (3x3 and 5x5) resulted in higher MSE and longer training times. Therefore, the 2x2 kernel size was selected as the most optimal option.

**Re: Imputation**
For CNN training, datasets cannot contain missing values. Several methods can address missing data, such as creating masks, removing rows/columns with NaNs, or imputing NaNs (e.g., using nearest neighbor cells). The AOD values from MODIS had approximately 30% missing values at random locations and 20% missing for the mean between Aqua and Terra (see left panel in Figure 10/A6). Since CNNs perform better with more data, imputation was selected as the preferred solution. This ensured sufficient data for training without compromising the accuracy of the final predictions. However, all results were calculated using un-imputed MODIS data.

**4. The limitation to midday**

The limitation to midday predictions is important and should be mentioned in the abstract, as there does not seem to be an easy solution within this framework to go beyond this, e.g. to provide data with shorter time steps or daily averages. This, combined with the fact that the model does not produce dust-specific data such as dust concentrations or emissions, limits the value of the model compared to the expectations raised by the present title and abstract.

We thank the reviewer for this comment.

We have amended the abstract to include the following wording:

Line 22: "24-hour (1-step) ahead predictions."

**Minor comments:**

**1. Page 3, Fig. 1**

Page 3, Fig. 1, caption: The term "dimensionality" seems inappropriate here, as it is used in the context of extending and shrinking of existing dimensions

We agree with the reviewer. Indeed, in this context, we use reshaping to extend and shrink existing dimensions.

Page 4: 'Therefore, we have replaced the term 'dimensionality' with 'input size' in Fig. 1 caption.

**2. Page 3, bottom:**

Page 3, bottom: Here only the climatological mean is mentioned as the baseline model, but clearly the persistence forecast mentioned on page 15 is both a simpler and better baseline model and should therefore be the reference.

We are gratefull to the reviewer for bringing this point to our attention.

We have consistently used the climatological mean as the baseline throughout the manuscript, as it provides a comparable reference to the CAMS model, which uses climatology as input states. As the difference between the MSE of climatology and persistence was minimal (0.00031), we have decided to be consistent and use climatology.

**3. Page 7, Fig. 3:**

Page 7, Fig. 3: Since all rows show AOD, I suggest using the same colour gradients

We thank the reviewer for this suggestion.

We have tried to apply this same colour gradients, however this choice did not make all relevant effects maximally visible.Thus, we have opted-in to use varied, colour-blind friendly palettes which were most effective in highlighting the differences between the conditions.

**4. Page 8,**

Page 8, 2nd paragraph: Please expand DOY

We thank the reviewer for spotting this typo.

Line 216: 'We have now expanded the 'DOY' to read 'Day of Year'.

**5. Page 9, Fig. 4**

Page 9, Fig. 4, caption: "The background image, ..." refers to the image actually shown in Fig. 11/S7

We are grateful to the reviewer for spotting this misplaced sentence.

We have now removed the entire last sentence from the Fig. 4 caption:

Page 9: "The background image, …'

**6. Page 10,**

Page 10, 1st paragraph: Surface soil moisture is another factor controlling dust emissions

We thank the reviewer for this suggestion.

Line 260: We have added 'soil moisture' to the $4^{th}$ sentence of the $3^{rd}$ paragraph

**7. Page 11,**

Page 11, "Timestamps": The meaning of "multiplied the file" is somewhat clear, but perhaps "expanded the tensor" or something like that would be more technically correct.

As suggested by the reviewer, we have reworded the sentence as follows:

4.2.3 Timestamps

$2^{nd}$ sentence:

Lines 343-346: 'We then expanded the array dimensions through replication to match the exact spatial resolution of atmospheric variables, resulting in a coverage of 31 x 51 grid cells for each day.'

**8. Page 12,**

Page 12, "ERA5 regridding": "vertical resolution" should read "horizontal resolution"

We thank the reviewer for bringing this typo to our attention.

Line 384: We have corrected 'vertical' to 'horizontal'.

**9. Page 13,**

Page 13, Eqs. (2), (3): Please define t as day of year.

We thank the reviewer for pointing out this omission. We have added the following wording to the end of the equations:

Line 401: ', where t represents the day of the year.'

**10. Page 13, tvt**

Page 13, "Training, validation, test split": You may add that splitting with consecutive time steps also avoids autocorrelation between the three subsets.

We thank the reviewer for this valuable suggestion. Therefore, we have added the following as a third and fourth sentence:

Lines 407-410: 'The use of consecutive time steps ensures that each subset is composed of data points that are temporally distinct. This reducies the risk of autocorrelation and improves the model's ability to generalize to new, unseen data.' [5]

**11. Page 13, last**

Page 13, last paragraph: Given that you assessed performance based on MSE, it is not surprising that optimising with MSE-Loss performs best, making the test of different loss functions not very meaningful.

We agree with the reviewer's comments raised above;

Lines 427-436: we have removed the references to these metric tests from our manuscript.

**12. Page 15,**

Page 15, "U-NET": "pool size 1 x 1" is confusing. What dimension are you downsampling, does this layer have any effect?

We thank the reviewer for spotting this point, as we agree that a pool size of 1 x 1 is indeed unusual and can be confusing.

Indeed, a pool size of 2 x 2 is more standard and intuitive, and it will reduce the spatial dimensions by half.

Thus, we have amended the U-NET architecture to use a pool size of 2 x 2, and we have updated the model description, the code in the ZENODO archive, as well as the training time and MSE in Table 2. Model performance and main conclusions were not affected by this change.

Line 480: we have amended the pool size reference form 1 x 1 to 2 x 2.

**13. Page 15,**

Page 15 and Page 3, Fig. 1:, "... while increasing the amount trainable parameters.": Do you mean increasing the number of channels/filters?

We thank the reviewer for this comment, as we agree that this part could have been explained more clearly.

In this particular instance both are true. The trainable parameters and channels increase when 2 x 2 kernel is combined with 2 x 2 stride in Conv2D layer. Here, the channels increased to 256 while trainable parameter increased by 262,400. Thus, we have expanded the reference of increasing input size to include the fourth dimension of channels.

Lines 497-498: Also, for clarity, we have replaced the reference to 'increasing parameters' with 'channels' in the model's description.

**14. Page 16,**

Page 16, Table 2: The persistence forecast is the more relevant baseline model and should be used in the table.

We thank the reviewer for raising this point.

Although it is true that persistence can be more intuitive in certain cases, we use climatology here for consistency reasons. This metric was used throughout the manuscript for comparisons with CAMS forecasts, which use monthly climatology as input for its states.

15. Page 28,

Page 28, Fig. 6: It is difficult to identify the different distributions, it would be easier without colouring the areas under the curves.

Page 30: We agree with the reviewer's comment and thus have changed Fig. 6 (now Fig. A2) to a density curve without shading.

16. Page 34,

Page 34, Fig 13: Why is the RMSE between the constant climatological mean and the varying observations constant throughout the year?

We grateful to the reviewer for spotting this omission, as the constant climatological mean was added in error.

Page 37: We have now removed the climatological baseline from the top panel of Fig. 13 (now Fig. A9).

References:

[1] Klingmüller, K. and Lelieveld, J.: Data-driven aeolian dust emission scheme for climate modelling evaluated with EMAC 2.55.2, Geosci. Model Dev., 16, 3013–3028, https://doi.org/10.5194/gmd-16-3013-2023, 2023.

[2] Kanngießer, F., & Fiedler, S. (2024): "Seeing" beneath the clouds—Machine-learning-based reconstruction of North African dust plumes. AGU Advances, 5, e2023AV001042. https://doi.org/10.1029/2023AV001042

References to comments responses:

[3] Wei, X., Cui, Q., Ma, L., Zhang, F., Li, W. and Liu, P., 2024. Global aerosol-type classification using a new hybrid algorithm and Aerosol Robotic Network data. Atmospheric Chemistry and Physics, 24(8), pp.5025-5045. https://acp.copernicus.org/articles/24/5025/2024/

[4] Francois Chollet, Deep Learning with Python: 12.5.2 A bag of tricks, Second Edition , Manning, 2021.

[5] Rasp, S., Dueben, P.D., Scher, S., Weyn, J.A., Mouatadid, S. and Thuerey, N., 2020. WeatherBench: a benchmark data set for data-driven weather forecasting. *Journal of Advances in Modeling Earth Systems*, *12*(11), p.e2020MS002203.

Cite Referee #1

Citation for this Referee comment: https://doi.org/10.5194/egusphere-2024-2259-RC1

**Referee #2 - Narendra Ojha**

This study presents a convolutional neural network (CNN) based model utilising ERA5 meteorological reanalysis and MODIS observations for simulating AOD over Saharan region in Africa. The CNN-based model (called as the DustNet) is shown to have substantially improved performance at very low computational costs when compared to a process-based model CAMS. Such applications (and evaluations) of novel approaches can be useful to represent accurate aerosol loadings (and thereby correcting their potential impacts on radiation/meteorology). However, several comments need to be addressed and the discussions need to be strengthened before the publication of the manuscript in GMD, as elaborated below:

**Major comments**

**1. Abstract**

Abstract is qualitative which requires improvement. It hardly informs the reader what model is used (a CNN), what datasets model has been built upon (ERA5 meteorology & MODIS-AOD), and how well (quantitatively) it performed over the study region.

We thank the reviewer for suggesting this improvement, with which we agree.

We have now amended the abstract to read:

Line 22-34: Suspended in the atmosphere are millions of tonnes of mineral dust that interact with weather and climate. Accurate representation of mineral dust in weather models is vital, yet it remains challenging. Large-scale weather models use high-power supercomputers and take hours to complete forecasts. Such computational burdens allow them to include only monthly climatological means of mineral dust as input states, inhibiting their forecasting accuracy. Here, we introduce DustNet, a simple, accurate, and super-fast forecasting model for 24-hour (1-step) ahead predictions of aerosol optical depth (AOD). DustNet is a custom-built 2-D Convolutional Neural Network (CNN) equipped with transposed convolution layers. The model is trained on selected ERA5 meteorology and past MODIS-AOD observational data as inputs. Our design of DustNet ensures that the model trains in less than 8 minutes and creates predictions in 2.1 seconds on a desktop computer, without the need to utilize any Graphics Processing Units (GPUs). Predictions created by DustNet outperform the state-of-the-art physics-based model at coarse 1° x 1° resolution at 95% of grid locations when compared to ground truth satellite data. The test results show that the daily mean AOD over the entire area highly correlates with MODIS observational data, with Pearson's $r^2$ = 0.91. Our results demonstrate DustNet's potential for fast and accurate AOD forecasting, which can easily be utilized by researchers without access to supercomputers or GPUs.'

**2. The study**

The study systematically discusses predictions of AOD and not of dust (as such). Although AOD in the study region is governed heavily by dust, there are discussions on impacts of intense fires also in the manuscript. This needs to be rectified. The methodology using MODIS observations may actually work for other regions also (as authors also suggest) where dust may or may not be the dominant factor.

We thank the reviewer for this informative and important comment, as it made us realize that our communication of methods for CNN model training should be improved. We have therefore amended the mentions of biomass burning and clarified within the text that DustNet was not specifically trained to pick up any other AOD particles, such as sea salt or black carbon. The paragraphs below were added to clarify that point:

**4. Methods**
**4.1 Study Area**

Lines 304-315: "This choice allowed us to gain a sufficient amount of training data, with 51 x 31 grid cells providing 1,581 pixels for each training day, thereby ensuring robust model performance. By selecting this region, we were able to strike a balance between training efficiency, training speed, and prediction accuracy, making it possible to achieve effective dust aerosol forecasting. Furthermore, this approach enabled us to train the model on a traditional desktop without relying on cloud resources for data storage, making our approach more accessible and cost-effective. Additionally, the study region effectively captures dust aerosol generation and transport on selected features, which is essential for accurate forecasting. Finally, by minimizing the area to the Saharan Desert and consequently reducing the amount of chosen

training features, we were able to avoid adding different ocean and terrain processes, leading to reduced model complexity without compromising performance."

**3. Discussion**

**2nd Paragraph:**

Lines 249-255: "The specific DustNet model architecture may be adapted for predicting other atmospheric particles. However, this would require retraining the model using input features that represent the chosen particle. For example, to capture aerosols due to black carbon, features such as land cover types, vegetation, leaf area index, and forest fire locations should be considered. Similarly, when aiming to capture atmospheric aerosols due to sea-salt particles, features including wave height, energy flux into waves, peak wave period, and ocean surface stress should be taken into account."

**3. Features**

A key area for improvement is analysing the model performance in terms of feature-importance. Out of so-many (>40) parameters considered in the model, discuss and highlight which features had the greatest impacts in controlling variability in the AOD.

We thank the reviewer for raising this important consideration. We agree that identifying the most important features that drive the model predictions is a clear area for improvement. Indeed, model interpretability is a key area of active research in convolutional neural networks and other AI models in general.

Therefore, we have followed a perturbation-based method for assessing feature importance [5, 6].

Page 41: We have added the resulting figure as supplementary Fig. 17 (A13) and included a relevant description as an additional subsubsection of statistical analysis:

**4.5.6 Feature Importance**

Lines 606-617: "We assessed feature importance using a perturbation-based method, where individual input channels were systematically altered to evaluate their contribution to model predictions. Specifically, each feature was zeroed out in turn, and the mean squared error (MSE) between the full prediction and the prediction with the altered input was calculated. This approach quantifies the sensitivity of the model's output to the absence of each feature, with higher MSE indicating greater importance. Results shown in supplementary Fig. A13 demonstrate that input channels corresponding to the features 'AOD 1 day lag' and 'vertical velocity at 850 hPa' exhibited the largest impact, indicating their relative importance for model predictions. Perturbation-based methods, such as this one, are widely used for assessing feature relevance in machine learning models due to their simplicity and interpretability (Covert et al. (2021); Molnar (2022))." [6, 7]

**4. Boundaries**

Page 4: Performance of spatial forecast: authors point out limitation of the DustNet near its domain boundaries (interestingly there CAMS performed better than DustNet). It is not clear why DustNet was not set up to cover a larger area to verify (and minimize) the suggested influence of boundaries. Not fixing this issue is confusing as the required datasets ERA5 and MODIS are readily available for larger region and the DustNet model is suggested to train (and run) quite fast.

We thank the reviewer for raising this important consideration.

The study area was carefully chosen to capture the dust aerosol generation and transport from the Sahara

Desert. Expanding the study research area to the North and South, where the predictions were lacking, would expand the study region into the Mediterranean Sea and Congolian rainforest, where mineral dust is not the dominant contributor to AOD. Including such areas could dilute the focus of the study and introduce confounding factors. Consequently, the objective of this particular study would change. Likewise, the research was performed on a traditional desktop computer; therefore, additional data would require extra resources, especially during the data wrangling process.

We do, however, recognise the need to communicate this point better. Thus, we have added the following improvements to our Discussion section:

**3 Discussion**

**4th paragraph:**

Lines 266-275: "Additionally, DustNet's predictions at the northern and southeastern locations of the region boundaries are visibly weaker than at the center (Fig. 2c and 2d). Since the predominant transport directions of the atmospheric dust during this study are confirmed as west and southwest (Fig. 3, especially 3b and 3c), which indicates that the northern and southeastern areas may be governed by processes not included in the feature selection of this study. This is not surprising, since the Mediterranean Sea is directly to the north of our study region, while the Congolian rainforest covers grids directly to the south and southwest of the boundaries. These indicate the potential for more skillful forecasts with a broader study area, which, together with additional features, could capture more nuanced processes above the oceans and rainforests."

**5. Daily data**

There has been a significant emphasis on the need of simulating AOD over daily scales (instead of using monthly means). It would be useful to draw a map of temporal correlation using data at daily resolution. It will tell clearly over which regions model better reproduces day-to-day variation (and not only the mean as shown currently in Figure 2). This will be more in line with the discussions in introduction.

We thank the reviewer for indicating this area of research improvement. We have amended all related figures to clearly state that they were drawn using data at a daily resolution. The only exception is Fig. 3, which shows annual and seasonal means; however, these were also calculated from daily data.

The values in Fig. 2 represent the mean RMSE calculated from daily data (n=1095) over three years (2020-2022) for predicted values at each location. For clarification, we have added the following as a second sentence in the Fig. 2 description:

Page 5: "Results for 24-hour (1-step ahead) predictions of daily AOD values (mean across the daily prediction time 2020-2022, n=1095) compared with the ground truth data from MODIS."

Page 38: For completeness, we have also added a supplementary Fig. 14 (A10), which shows a correlation map between DustNet and MODIS, and CAMS and MODIS. Here again, the correlation was calculated using daily data from the prediction period (2020-2022) with n=1095 for each location grid.

A paragraph below was also included in the supplementary material to describe the figure above:

Lines 914-924: "Represented in Figure A10 is the daily correlation coefficient between the ground true values from MODIS and DustNet predictions in panel a), and between MODIS and CAMS in panel b). The area of the Saharan Desert, where mineral dust is the main contributor to the AOD values, is the stronghold of DustNet-produced predictions. In comparison, CAMS displays a lower (weaker) correlation with MODIS

over areas in the Saharan Desert, which were highlighted in previous figures (Fig. 3 and Fig. A6) as dust generation regions. The regions south of 8°N and east of 10°E display weaker correlation for both models. This is not surprising, since DustNet's training regime did not include any data indicative of black carbon or secondary organic aerosols, which seasonally dominate the AOD over the equatorial region of Central African forests (Jo et al. (2023))." [8]

**6. Fig 3 Caption:**

Fig 3: Caption: Enhancement in AOD due to biomass-burning is also captured. How the DustNet segregates influences of dust vs fire-emissions, it only tunes AOD? Check the entire caption carefully ("CAMS represent biomass-burning related AOD"), also seems coming direct. How is that analyzed? The caption is really a long paragraph. Better to only describe the figure in caption and make a detailed discussion in the relevant sections clarifying these points.

We thank the reviewer for this important comment.

We have now amended the Figure 3 caption and removed the references to biomass burning.

Page 7: **"Fig. 3** Annual and quarterly means of daily AOD values for 2020-2022. All mean AOD values were calculated from daily 24-hour ahead predictions. The **left** column represents AOD values from MODIS observations, predictions from DustNet are in the **middle**, while forecasts from CAMS are in the **right** column. **Row a)** compares the 3-year annual mean AOD between the observations and models. In **row b)**, the 3-year mean of daily AOD for Q1: January - March is shown, noting the main generation site of the Bodélé Depression (dark blue) and the southwestward transport of mineral dust. In **row c)**, these same means are shown but for Q2: April - June. **Row d)** shows that both models, CAMS and DustNet, skillfully detected the northward shift of mean AOD transport during Q3: July - September. In **row e)**, the seasonal decrease in aerosol activity for Q4: October - December is skillfully captured by both models when compared to observations from MODIS. Note here the change in the colour bar range."

**7. Fig 4, S1 & S4**

Fig 4 and related analysis and discussion needs re-check. As of now, here the correlation between CAMS and MODIS is reported to be 0.01 which is too low (bit strange). See for example another figure (Fig S1) where the same two datasets correlate nicely (r2 = 0.6). Is this contrast due to averaging of days across years? It will be more appropriate to evaluate on day-to-day data over 3 years continuous time-series. (This revision will not change overall conclusion as DustNet's performance seems clearly superior with r2 = 0.9).

We thank the reviewer for bringing this misscommunication to our attention.

**Regarding the difference between Fig. 4, S1, and S4:** The observed differences between the results in Fig. 4 and supplementary Fig. S1 are due to spatial averaging. Supplementary Fig. S1 shows mean values across the entire study area (mean over the Sahara Desert), where overall, CAMS and MODIS correlate well.

In contrast, Figure 4 (and the corresponding supplementary Fig. S4) shows values for specific 4 grid locations extracted from the study area: Bodélé Depression, Kano, Nouadhibou, and Gulf of Guinea (one grid point for each location). For clarity, we have amended the first caption sentence of supplementary Fig. S1 (now Fig. 5/A1), and Fig. S4 (now Fig. A4) which now read:

Page 7: "Fig. A1: Spatially averaged daily AOD (2020-2022, n=1095) regressed between model predictions and MODIS data."

Page 32: "Fig. A4: Scatter plot relationship between predicted **daily** AOD values (2020-2022) and MODIS

data."

**Regarding day-to-day data:** As suggested by the reviewer, we have added a supplementary Fig. 18 (A14) to aid examination of day-to-day variations.

Page 42: The Fig. 18 (A14) shows the results for each prediction day at the selected four grid locations.

**8. Discussion**

Discussion: "Despite DustNet not being trained….dust generation, seasonal variations….skilfully represented" is not convincing. Model is actually trained on long-term meteorological data so is exposed to seasonal variations. Regarding dust, it is not simulating that. It is simulating AOD variations on which it is trained explicitly. So these arguments need to be revised.

The reviewer is raising a valid point for which we are grateful. Thus, we have rephrased the third sentence of the 'Discussion and Conclusions' section to read:

Lines 242-244: "DustNet also skillfully captures aspects of atmospheric processes such as dust generation, transport, and seasonal variations when compared to the satellite data."

Please also see the second paragraph (lines 249-255) added to the Discussion section which is included in response #2.

**Other Comments:**

**1. Abstract**

Abstract last line: "transform our understanding of dust's impacts on weather patterns". I did not find any significant discussion on this aspect inside the paper.

We thank the reviewer for this important scrutiny.

We have amended the last sentence of the abstract to read as follows:

Lines 32-34: "Our results demonstrate DustNet's potential for fast and accurate AOD forecasting, which can be easily utilized by researchers without access to supercomputers or GPUs."

**2. 1000 hPa data**

Data at 1000 hPa has been used as a feature. Check if there are any significant variations in the topography and then sometimes 1000 hPa may fall below surface pressure in ERA5?

We appreciate the reviewer's comment, which highlighted the omission of our terrain data source.

The topographic data utilized in this study were obtained from the Joint Institute for the Study of the Atmosphere and Ocean (JISAO) and did not exhibit significant variations.

Furthermore, all 43 features are now included in the Supplementary Figure 17 (A13).

**3. BLH**

Are variations in boundary layer height (BLH) considered in the model which can perturb aerosol distribution / vertical mixing.

We appreciate the reviewer for raising this point; however, we have not considered the Boundary Layer Height (BLH) as an input feature.

**4. Vertical res**

The "vertical resolution of 0.25° × 0.25°" on Page 12 likely refers to horizontal resolution.

We are thankful to the reviewer for spotting this ommision.

Line 384: We have now replaced "vertical" with "horizontal".

**5. GPU info**

Page 13: "….Macbook Pro….32 GB RAM". Probably giving number of CPUs/GPUs utilized will be useful.

We appreciate the reviewer for raising this point, as it will provide an important addition to our manuscript.

We have amended the main text to include information on GPU usage, as follows:

Lines 427-428: "The models did not utilize any GPUs and can thus be replicated by users without access to a supercomputer."

**6. Overfitting**

Discuss how you ensured that model is not overfitted. The model has more than 40 features but training sample size is a few thousand. Such feature-to-sample ratio may, in some cases, lead to overfitting. Have you analyzed plot of training and validation loss (or accuracy) over the epochs. If necessary, features could be reduced using suitable feature selection techniques.

We appreciate the reviewer for raising this question.

We are kindly pointing to the supplementary Fig. 12 (now Fig. A8), which illustrates the training versus validation performance over the epochs.

The final version of the DustNet model was equipped with a custom recall mechanism for 'early stopping,' with 'patience' set to 4 and the 'save best only' option set to 'True.' Consequently, the model automatically halted training when the validation loss did not improve for four consecutive epochs, and only the best-fit model was saved for analysis.

Furthermore, the statistical analyses were performed on test data to which the model did not have access during training. Therefore, if the model were to overfit, the results on unseen data would not have been as reliable as presented.

References to comments responses (Referee #2):

[6] Covert, I., Lundberg, S., and Lee, S.-I.: Explaining by removing: A unified framework for model explanation, Journal of Machine Learning
Research, 22, 1–90, 2021.

[7] Molnar, C.: Interpretable Machine Learning, chap. Chapter 10: Neural Network Interpretation, Github, 2 edn., https://christophm.github.io/
interpretable-ml-book, 2022.

[8] Jo, D.S., Tilmes, S., Emmons, L.K., Wang, S. and Vitt, F., 2023. A new simplified parameterization of

secondary organic aerosol in the Community Earth System Model Version 2 (CESM2; CAM6. 3). *Geoscientific Model Development Discussions*, *2023*, pp.1-24.

---

## Author Response (AR3)

Saturday, 22 February 2025

[Figure]

Skillful neural network predictions of Saharan dust
Trish E. Nowak, Andy T. Augousti, Benno I. Simmons, and Stefan Sieger
Pre-print: https://doi.org/10.48550/arXiv.2406.11754

Dear Prof. Tost and the Editorial Team,

We would like to thank you for the comments included in this round of the revision process. These will inadvertently improve the quality of our manuscript.

Please find below our point-by-point response to your suggestions, along with a list of all relevant changes made.

We trust that these revisions adequately address your comments and contribute to an enhanced, more clearly organised manuscript. We appreciate your constructive feedback and the opportunity to improve our work.

Thank you for your time and consideration.

Sincerely,

*Trish Nowak - on behalf of all the authors*
*PhD Candidate*
*UKRI CDT in Environmental Intelligence: Data Science & AI for Sustainable Futures*
*University of Exeter*
*https://mathematics.exeter.ac.uk/staff/pn284*

**Editors comments:**

Dear authors,

Even though the reviewers comments are mainly answered, and some changes to the text are included, I think that the manuscript needs a substantial restructuring, i.e., at the moment the Methods Section comes after the Results and Discussion. This might be a typical style for a publication in Nature, but does not follow the typical GMD publication style. Consequently, please change the manuscript layout, such that it contains after the Introduction a section on Data and Methods. Afterwards please present your results; this also includes a better separation of what is a result and what is a method. This does not become clear in the current manuscript style. Furthermore, some parts of the Methods section also include some discussions such that also here the separation of pure methodology versus discussion of results needs to be sharpened.

At the moment, the manuscript is missing a Conclusion / Summary section. As some readers would like to get the main message of the paper only, please provide such a section.

After the structural and conceptual overhaul of the manuscript a publication might become possible.

**Authors response:**
We would like to kindly thank the editors for their constructive comments. Below we have included the point-by-point responses to each of the comments raised.

1. **Manuscript Layout and Structure**

   ○ **Change Implemented:** We have restructured the manuscript so that after the Introduction, a dedicated "Data and Methods" section now follows. This aligns the format with the typical GMD publication style, as indicated in the GMD's LaTeX template.

2. **Separation Between Methods and Results**

   ○ **Change Implemented:** We have revised the Methods section to exclusively detail the data acquisition and analytical procedures. Any discussions or interpretations that were previously included with the methods have now been moved to the Results section.

3. **Addition of a Conclusion/Summary Section**

   ○ **Change Implemented:** We have added a new Conclusions section at the end of the manuscript. Here, we succinctly outline the main findings and implications of our study. This should provide a clear takeaway for readers that seek the core message of the paper.

4. **In-Text Citation Formatting**

   ○ **Change Implemented:** All in-text citations have been reformatted to follow the prescribed GMD style (name, followed by date) instead of using numerical citations. We ensured the compliance with GMD's submission guidelines as detailed on the journal's website and the LaTeX manuscript template.

5. **Supplementary Material**

   ○ **Change Implemented:** The supplementary material was extracted from the main manuscript file and uploaded as a separate .zip file. In addition, the file has been renamed according to GMD's standards.